# Structural basis for the recognition of sulfur in phosphorothioated DNA

Guang Liu [1,2], Wencheng Fu[1], Zhenyi Zhang[1], Yao He[3], Hao Yu[1], Yuli Wang[1], Xiaolei Wang[1], Yi-Lei Zhao [1], Zixin Deng[1], Geng Wu[1] & Xinyi He [1]

There have been very few reports on protein domains that specifically recognize sulfur. Here we present the crystal structure of the sulfur-binding domain (SBD) from the DNA phosphorothioation (PT)-dependent restriction endonuclease ScoMcrA. SBD contains a hydrophobic surface cavity that is formed by the aromatic ring of Y164, the pyrolidine ring of P165, and the non-polar side chains of four other residues that serve as lid, base, and wall of the cavity. The SBD and PT-DNA undergo conformational changes upon binding. The $S^{187}RGRR^{191}$ loop inserts into the DNA major groove to make contacts with the bases of the $G_{PS}GCC$ core sequence. Mutating key residues of SBD impairs PT-DNA association. More than 1000 sequenced microbial species from fourteen phyla contain SBD homologs. We show that three of these homologs bind PT-DNA in vitro and restrict PT-DNA gene transfer in vivo. These results show that SBD-like PT-DNA readers exist widely in prokaryotes.

[1] State Key Laboratory of Microbial Metabolism, School of Life Sciences & Biotechnology, The Joint International Research Laboratory of Metabolic & Developmental Sciences, Shanghai Jiao Tong University, 200240 Shanghai, China. [2] State Key Laboratory of Bioreactor Engineering, East China University of Science and Technology, 200237 Shanghai, China. [3] Imaging and Characterization Core Lab, King Abdullah University of Science and Technology, Thuwal 23955-6900, Saudi Arabia. Correspondence and requests for materials should be addressed to G.W. (email: geng.wu@sjtu.edu.cn) or to X.H. (email: xyhe@sjtu.edu.cn)

DNA modifications, such as methylation and hydroxymethylation, play important roles in regulating gene expression in both prokaryotes and eukaryotes. In vertebrates, methylation and hydroxymethylation of particular DNA bases, for example, 5-methylcytosine and 5-hydroxymethylcytosine, are related to genomic imprinting, embryogenesis, and oncogenesis. On the other hand, DNA methylation and hydroxymethylation in bacteria usually function as modification-restriction mechanisms to prevent invasion of foreign DNA from other bacteria or phages[1].

In contrast to methylation, DNA phosphorothioation is unique because it occurs on the DNA phosphodiester backbone rather than its bases[2]. In natural PT-DNA, a non-bridging oxygen atom in the $R_p$ configuration on the phosphodiester bond is replaced with sulfur by Dnd proteins[3]. DNA PT modification is widely present in more than 200 different bacteria and archaea[4–6]. To date, four consensus sequence patterns of PT modifications have been identified, with $G_{PS}GCC/G_{PS}GCC$ for *Streptomyces lividans* 1326[7], $G_{PS}AAC/G_{PS}TTC$ for *Escherichia coli* B7A[5], $G_{PS}ATC/G_{PS}ATC$ for *Bermanella marisrubri* RED65, and $C_{PS}CA$ for *Vibrio cyclitrophicus* FF75[5] as a few examples. A recent study revealed shared consensus sequences for PT-methylated and $^{6m}A$-methylated DNA, suggesting a coevolution of different DNA modifications[8]. The reported biological functions of DNA PT modification include conferring resistance to oxidation to the host bacteria[9–13], restricting gene transfer among different bacteria[14,15], and influencing the global transcriptional response[16,17], among others.

DNA modifications are usually recognized by specific reader proteins that mediate the biological functions encoded by the modifications. For example, the SET and RING-associated (SRA) domain specifically recognizes 5-methylcytosine modification in both prokaryotes and eukaryotes[18–27] and the bromodomain functions as a specific reader for acetylated lysines on histones[28]. In comparison with methylation on paired bases that are embedded either in the axial center or internal grooves, phosphorothioation on the external edge of the DNA double-helix allows a better accessibility for receptor proteins, and furthermore enables a pathway of crosstalk with neighboring sequences based on a recent theoretical investigation[29]. The Type-IV restriction endonuclease (REase)ScoA3McrA (abbreviated as ScoMcrA hereafter) catalyzes multiple double strand cleavages 16–28 nucleotides (nt) away from the phosphorothioate linkage with the core sequence $G_{PS}GCC/G_{PS}GCC$. Moreover, ScoMcrA also cuts Dcm-methylated DNA 12–16 nt away from the methylation site[14,15]. As either modification is sufficient to elicit cleavage, it has been postulated that more than one recognition domain of ScoMcrA could be involved in the discrimination of modified PT-DNA from normal DNA[30].

In addition to the different localization in the DNA structure compared with base methylation, PT modification also introduces a sixth element, sulfur, into the DNA phosphodiester backbone. Sulfur is a constituting element for a variety of biomacromolecules and cofactors, such as methionine, cysteine, thiamine, biotin, thiouridine, lipoic acid, iron–sulfur cluster, etc. Several structures of protein in complex with sulfur-containing molecule have been reported and their residues coordinated with sulfur have been identified. In the structure of *E. coli* methionyl-tRNA synthetase, the $S^\delta$ atom of L-methionine interacts with Y260 and L13[31]. In S-adenosyl methionine synthesis, the sulfur of methionine is coordinated by a magnesium ion that is salt-bridged with O5′ from ATP[32]. In the sulfur transfer process from L-cysteine to various acceptors, such as iron–sulfur clusters, persulfide groups are usually formed between sulfur and cysteine residues in the active center[33,34]. However, there is no previous report of a specific protein domain recognizing a sulfur atom on

biological molecules. The DNA PT-dependent REase ScoMcrA distinguishes PT-DNA from normal DNA which differs only by one oxygen-to-sulfur swap, and presumably encodes a domain to specifically recognize sulfur on PT-DNA.

Here, in contrast to the previous postulation that the salt-bridge between the negatively charged phosphorothioated sulfur and the positively charged amino group of lysine/arginine provides primary interaction between PT-nucleotide and protein[35], we reveal a highly conserved non-polar sulfur-recognizing surface cavity in SBD by determining the crystal structure of ScoMcrA −SBD in complex with PT-DNA at 1.70 Å resolution (Protein Data Bank (PDB) accession number 5ZMO [https://www.rcsb.org/structure/5ZMO]). This cavity is surrounded by a hydrophobic wall consisting of the methylene groups from H116, R117, Y164 and the methyl group from A168, and the bottom of the cavity is formed by the pyrolidine group of P165. Besides, R117 (lysine in most homologs) forms salt bridges with the phosphorothioate oxygen and sulfur (also see the comparison and contrast of oxygen and sulfur atoms in Supplementary Discussion). The aromatic ring of Y164 serves as the lid of the sulfur-recognizing cavity, and is opened 98° outward when the sulfur atom is inserted into the cavity. Key residues that are directly involved in protein–sulfur and protein–base interactions were mutated and assayed for their contribution to PT-DNA binding. The SBD domain can be detected in more than 1000 sequenced species from 14 bacterial phyla. We purified three ScoMcrA−SBD homologs and verified that they distinguish $R_p$−PT-DNA from $S_p$−PT-DNA and normal DNA. Furthermore, their heterologous expression restricted transfer of foreign *dnd* gene cluster in vivo. The molecular recognition mechanism of sulfur in PT-DNA by a highly conserved protein domain we present here will promote the understanding of the biological function of DNA phosphorothioation and sheds light on the recognition of other sulfur-containing molecules.

## Results

**Structure of full-length (FL) ScoMcrA protein.** The molecular mechanism of how PT-DNA is recognized has remained elusive until now. ScoMcrA is the first phosphorothioation-dependent REase, and it has been postulated to harbor a domain or motif that recognizes PT-DNA with the sulfur atom in the $R_p$ configuration (lane 3 in Fig. 1a, Supplementary Fig. 27), but not PT-DNA with the sulfur in the $S_p$ configuration (lane 4 in Fig. 1a) or non-PT DNA (lane 2 in Fig. 1a). In order to elucidate the molecular mechanism of ScoMcrA recognition of PT-DNA, we determined the crystal structure of the *Streptomyces coelicolor* FL ScoMcrA to 3.15 Å resolution (PDB accession number 5ZMM) using the single-wavelength anomalous dispersion (SAD) method with a selenomethionine (SeMet) derivative (Fig. 1b–d, Supplementary Fig. 1, Supplementary Table 1). In each asymmetric unit, there are six ScoMcrA molecules (Supplementary Fig. 2a), which are assembled into three dimers (Supplementary Fig. 2b). This is consistent with the observation that ScoMcrA behaves as a dimer in solution (Supplementary Fig. 2c).

Each ScoMcrA protomer consists of a head domain, an uncharacterized domain, a SRA-like domain, and an HNH domain. These four modular domains are arranged analogous to beads on a string (Fig. 1d), with the latter three domains all contributing to the dimerization contacts (Fig. 1c). The head domain is small, and with an unknown function (Supplementary Fig. 3a). Superimposition of the six ScoMcrA protomers in the asymmetric unit shows that the head domain exhibits substantial positional variance with respect to the other domains, suggesting that it is rather flexibly attached to the rest of the molecule (Supplementary Fig. 4). The uncharacterized domain binds

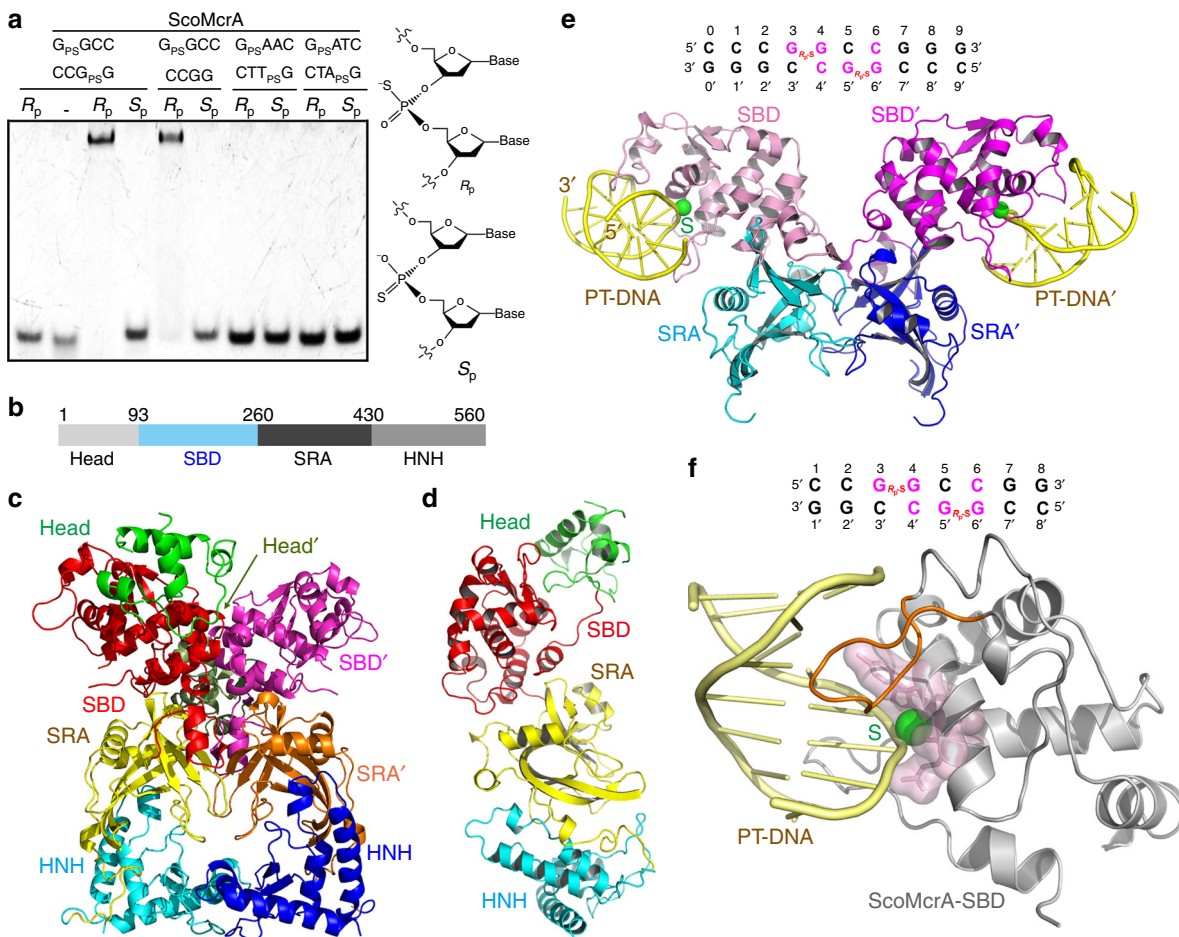

**Fig. 1** ScoMcrA employs its SBD domain to recognize PT-DNA. **a** *Streptomyces coelicolor* ScoMcrA specifically associated with PT-DNA and hemi–PT-DNA containing the $G_{PS}GCC$ core sequence, with the sulfur atom in the $R_p$, but not in the $S_p$, stereo-specific configuration. Lane 1 serves as a control with no protein added. **b** Domain organization of ScoMcrA. **c** Structure of the full-length ScoMcrA dimer. The four domains of ScoMcrA, head, SBD, SRA, and HNH domains, are labeled. **d** Structure of a ScoMcrA protomer. **e** Structure of ScoMcrA–SBD-SRA in complex with a 10 base pair PT-DNA. The DNA sequence is shown at the top, with the bases recognized by ScoMcrA highlighted in magenta. **f** Structure of ScoMcrA–SBD in complex with an eight base pair PT-DNA. Sulfur-recognizing residues are shown in stick as well in surface representations. The "S[187]RGRR[191] loop" recognizing the bases of PT-DNA is colored in orange

specifically to PT-DNA whose sulfur atom is in the $R_p$ but not in the $S_p$ configuration (Supplementary Fig. 5), therefore we named it sulfur-binding domain (SBD). It exhibits a novel fold consisting of 10 α-helices as the main body and a three-stranded β-sheet on the periphery (Supplementary Fig. 3b). DALI search revealed no significant structural homolog of ScoMcrA-SBD, with the most similar hit being the origin recognition complex subunit-2 (PDB code: 1W5S [https://www.rcsb.org/structure/1W5S], *Z* score = 8.7). Both the SRA and the HNH domains are structurally similar to known SRA domains[18–27] and HNH[36] domains, that are, respectively, responsible for recognizing 5-methylcytosines and performing double strand cleavage on DNA (Supplementary Fig. 3c,d). In the HNH domain, two β-strands and an α-helix harbor the active site residues H508, N522, and H531, that are critical for its endonuclease activity[14]. Four cysteine residues, C484, C489, C527, and C530, coordinate a $Zn^{2+}$ ion in a zinc finger-like structure (Supplementary Fig. 3d).

**Structure of the ScoMcrA-SBD domain in complex with PT-DNA.** After numerous crystallization trials using various ScoMcrA fragments and different lengths and sequences of PT-DNA oligonucleotides, we successfully determined the crystal structures

of the SBD-SRA domains of ScoMcrA (residues 91–442) in complex with a 10-base pair (bp) PT-DNA (5′-CCCG_PSGCCGGG-3′) at 3.30 Å resolution (PDB accession number 5ZMN; Fig. 1e), and the SBD domain of ScoMcrA (residues 91–260) in complex with an 8-bp PT-DNA (5′-CCG_PSGCCGG-3′) at 1.70 Å resolution (Fig. 1f and Supplementary Table 1). Similar to the FL ScoMcrA protein, ScoMcrA–SBD-SRA forms a dimer, with the SRA domain mediating dimerization but no association with PT-DNA (Fig. 1e). In contrast, the ScoMcrA–SBD structure exists as a monomer (Fig. 1f). Although there are sulfur atoms in both strands of PT-DNA, which contain a palindromic $G_{PS}GCC$ core sequence, only one sulfur atom in PT-DNA is recognized by ScoMcrA-SBD in both structures (Fig. 1e, f). In support of these structural observations, both FL ScoMcrA and ScoMcrA-SBD interacted with hemi-PT-DNA, which has only one of the two DNA strands phosphorothioated, as shown in EMSA assays (lane 5 in Fig. 1a and Supplementary Fig. 5).

**Recognition of the $R_p$ sulfur atom in PT-DNA by ScoMcrA-SBD.** The sulfur atom on PT-DNA fits snugly in a cavity on the surface of SBD (Fig. 1e, f and Supplementary Fig. 6). The wall of

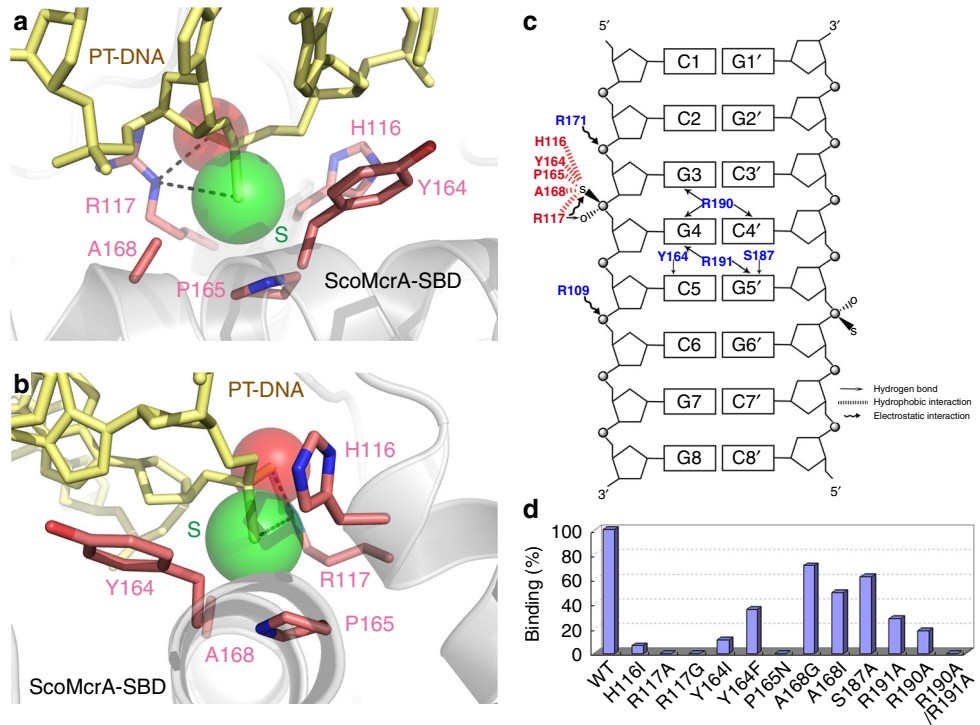

**Fig. 2** Residues interacting with the sulfur atom. **a** Close-up view of the sulfur-binding cavity on ScoMcrA-SBD. The sulfur atom of PT-DNA is recognized by the pyrolidine ring of P165, the β-methylene groups of Y164 and H116, the β-methyl group of A168, and the γ-methylene group of R117 by hydrophobic interactions, as well as by the guanidinium group of R117 through electrostatic interaction. Electrostatic interactions are indicated as magenta dashed lines. **b** The view in **a** was rotated 90° towards left. **c** Schematic summary of the interactions between ScoMcrA-SBD and PT-DNA. **d** Mutations of key sulfur-recognizing or base-contacting residues disrupted or decreased the association between ScoMcrA-SBD and PT-DNA as analyzed by the EMSA assay. The original EMSA gels are shown in Supplementary Fig. 9

this sulfur-binding cavity of SBD is lined with non-polar functional groups, including the β-methylene groups of Y164 and H116, the β-methyl group of A168, and the γ-methylene group of R117. The hydrophobic pyrolidine ring of P165 forms the bottom of this sulfur-binding cavity. In addition, the sulfur-binding cavity also contains positively charged element, such as the guanidinium group of R117 (Fig. 2a–c).

Compared to oxygen, the electronegativity of sulfur is substantially lower, being 2.58 compared to 3.44 for that of oxygen by the Pauling scale[37]. Hence, it can be expected that phosphorothioate is less hydrophilic and more hydrophobic than phosphate. In the PT-DNA bound ScoMcrA-SBD structure, the sulfur atom forms hydrophobic interactions with the pyrolidine ring of P165, the β-methylene groups of Y164 and H116, the β-methyl group of A168, and the γ-methylene group of R117 (Fig. 2a–c and Supplementary Fig. 7). Besides, the guanidinium group of R117 forms salt bridge with the phosphorothioate in PT-DNA[38] (Fig. 2a–c). In the $S_P$ configuration, the sulfur atom would be incorrectly positioned to be accommodated by the sulfur-binding cavity (Supplementary Fig. 8a, b). Moreover, the reverse stereochemistry of phosphorothioate would lead to an unfavorable interaction between guanidinium cation and phosphorothioate anion, in particular for the low charge on the sulfur atom and its larger van der Waals radius. This is consistent with our biochemical results that ScoMcrA or its SBD domain did not associate with PT-DNA with the sulfur atom in the $S_P$ configuration (Fig. 1a and Supplementary Fig. 5). Therefore, the SBD domain of ScoMcrA manipulates the subtle balance between hydrophobic and electrostatic interactions to recognize the sulfur atom in PT-DNA.

To verify our structural observations, we performed an EMSA assay to examine how various mutations of the sulfur-recognizing

residues of ScoMcrA-SBD affect PT-DNA binding. Replacement of the non-polar P165 in the center of the sulfur-binding cavity to hydrophilic residues, such as asparagine clearly caused no DNA shifting by ScoMcrA-SBD (Supplementary Fig. 9), indicative of its critical importance. Furthermore, mutating the positively charged R117 to neutral residues, such as alanine or glycine abolished its electrostatic interaction with the sulfur atom in PT-DNA and led to a complete loss of ScoMcrA-SBD/PT-DNA complex formation. Point mutations of H116I, Y164I, and A168I, which introduced extra γ-methyl groups to their side-chains, created steric hindrance between the sulfur atom of PT-DNA and the sulfur-binding cavity of SBD and PT-DNA binding of these mutants was reduced by 94%, 89%, and 51%, respectively (Fig. 2d and Supplementary Fig. 9). A168 is not as conserved as some other residues, such as P165 in ScoMcrA-SBD homologs (see Fig. 6a below) and it might contribute to PT-DNA binding through van der Waals interactions.

The binding affinities of these ScoMcrA-SBD mutants and PT-DNA were also measured by fluorescence polarization. It was confirmed that a mutation of P165 or R117 substantially disrupted the ability of ScoMcrA-SBD to bind PT-DNA, while mutations of other residues, such as H116 also diminished the association to varying degrees (Supplementary Fig. 10; since no other amino acid has the same backbone connectivity as proline, mutating a proline to another amino acid will change the backbone of the SBD protein and might lead to loss of its activity because of improper folding instead of changing of hydrophobicity). The EMSA and fluorescence polarization assay results indicate that both hydrophobic and electrostatic interactions, as well as correct folding of the SBD domain, play crucial roles for the sulfur-binding cavity of ScoMcrA-SBD to recognize the sulfur atom in PT-DNA.

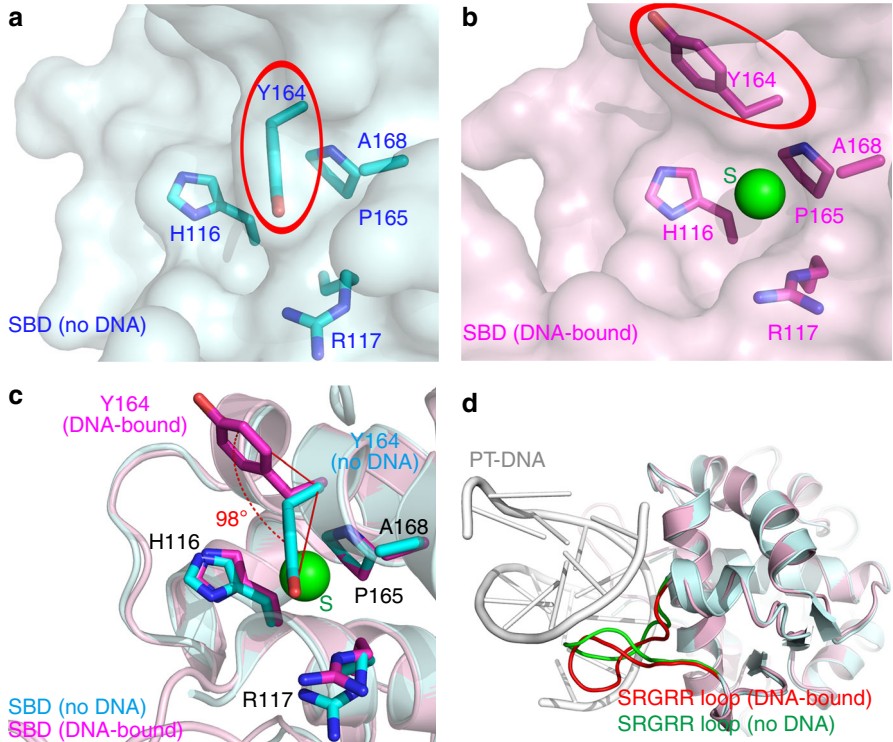

**Fig. 3** ScoMcrA-SBD undergoes conformational change upon recognition of PT-DNA. **a** In the PT-DNA-unbound state of ScoMcrA-SBD, the phenyl ring of the side-chain of ScoMcrA-Y164 covers the sulfur-binding cavity. **b** In the PT-DNA-bound state, the hydroxyphenyl group of Y164 is flipped open to allow the sulfur atom of PT-DNA to access the sulfur-binding cavity. **c** Comparing the PT-DNA-bound and PT-DNA-unbound states of ScoMcrA-SBD, the hydroxyphenyl group of Y164 is rotated 98° upon association of PT-DNA. **d** Association of PT-DNA also triggers a substantial conformational change of the "S[187]RGRR[191] loop" of ScoMcrA-SBD

**The S[187]RGRR[191] loop recognizes the G$_{PS}$GCC core sequence.** ScoMcrA specifically recognizes PT-DNA with the G$_{PS}$GCC core sequence[14] (Fig. 1a), which is one of the four known consensus PT-DNA sequences existing in bacteria. Our structure reveals that the tip of the α9–α10 loop of ScoMcrA, whose sequence is S[187]RGRR[191], inserts into the major groove of PT-DNA and provides base-specific contacts for the G$_{PS}$GCC core sequence. At the center of this interface, the guanidinium group of R190 in the S[187]RGRR[191] loop forms two hydrogen bonds with the O6 atoms of G$^3$ and G$^4$ of the G$^3_{PS}$G$^4$C$^5$C$^6$/G$^{6'}$G$^{5'}$C$^{4'}$C$^{3'}$ core sequence (′ denotes bases on the complementary strand). In addition, the guanidinium group of R191 forms two hydrogen bonds with the O6 atoms of G$^4$ and G$^{5'}$. Moreover, the hydroxyl groups of S187 and Y164 form hydrogen bonds with the N7 atom of G$^{5'}$ and the N4 atom of C$^5$, respectively. Furthermore, the N4 atom of C$^6$ forms a weak hydrogen bond with the hydroxyl group of Y164 and an electrostatic interaction with the carboxyl group of D160. Lastly, the positively charged R109 and R171 form electrostatic interactions with the negatively charged backbone phosphate of PT-DNA (Fig. 2c, Supplementary Figs. 11 and 12). The hydrogen bonds of Y164 to the N4 of C$^5$ and the N4 of C$^6$ atoms might contribute to flipping of the hydroxyphenyl group of Y164 and opening the sulfur-binding pocket of ScoMcrA-SBD. The importance of Y164 for the recognition of the G$_{PS}$GCC core sequence is consistent with the fact that it is not well conserved in ScoMcrA-SBD homologs as different PT-DNA core sequence motifs would require different corresponding residues for binding.

Consistent with these structural observations, substitution of R190 or R191 by alanine caused a 82% or 72% decrease in the association between ScoMcrA-SBD and PT-DNA, respectively, and double mutation of R190A/R191A entirely abolished the

complex formation between ScoMcrA-SBD and PT-DNA (Fig. 2d and Supplementary Fig. 9), which were confirmed by the fluorescence polarization assay (Supplementary Fig. 10). In addition, the single point mutation of Y164 to phenylalanine led to a 65% loss of complex formation between ScoMcrA-SBD and PT-DNA, and replacement of S187 by alanine reduced the binding affinity of ScoMcrA-SBD for PT-DNA by 38% (Fig. 2d and Supplementary Fig. 9).

**PT-DNA binding induces conformational change of ScoMcrA-SBD.** A comparison of ScoMcrA-SBD structures in the free and PT-DNA bound states revealed that PT-DNA binding induces a rotation of the side-chain of Y164 in the sulfur-binding cavity of ScoMcrA-SBD. When not in complex with PT-DNA, Y164 is flexible and samples a variety of conformations. Of the six ScoMcrA molecules in the asymmetric unit, only two have clearly observable electron densities for the side-chain of Y164 and both of which are in the closed states (Supplementary Fig. 13). In these two molecules, the hydrophobic phenyl ring of the side-chain of Y164 covers the opening of the non-polar sulfur-binding cavity (Fig. 3a and Supplementary Fig. 14a). The other four ScoMcrA molecules lack clearly observable electron densities for the side-chain of Y164 (Supplementary Fig. 13).

On the other hand, upon binding of PT-DNA, the hydroxyphenyl group of the side-chain of Y164 is flipped open allowing the sulfur atom in PT-DNA to access the ScoMcrA-SBD cavity (Fig. 3b and Supplementary Fig. 14b). Hence, the association with PT-DNA leads to a 98° rotation of the hydroxyphenyl group from Y164 side-chain (Fig. 3c). This conformational change of ScoMcrA-Y164 could either be caused by the binding of ScoMcrA-SBD to the sulfur atom of PT-DNA or by the binding

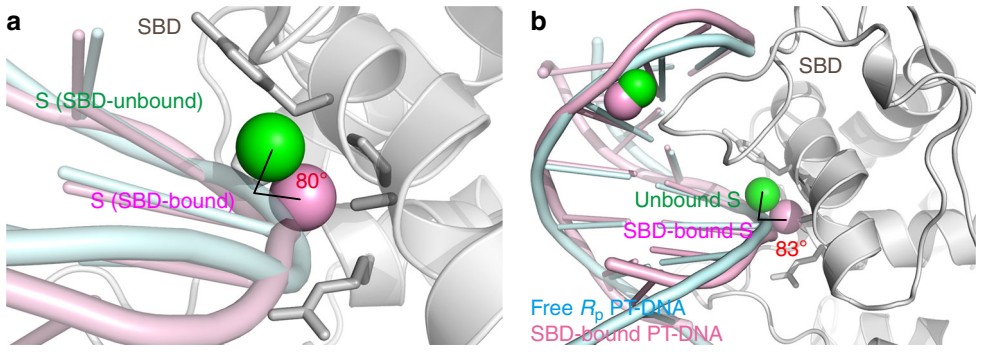

**Fig. 4** The sulfur atom rotates ~80° outward upon association with ScoMcrA-SBD. **a** Comparison between the SBD-bound and SBD-unbound strands of the same PT-DNA molecule in the structure of ScoMcrA-SBD in complex with PT-DNA. An 80° outward rotation of the sulfur atom can be observed, which makes it more suitable for accommodation by the sulfur-binding cavity on ScoMcrA-SBD. **b** Comparison between the free and SBD-bound PT-DNA also shows that the sulfur atom rotates 83° outward upon its recognition by ScoMcrA-SBD

to the GGCC motif, or both may make contributions. The $S^{187}RGRR^{191}$ loop of ScoMcrA also undergoes a substantial conformational change so that it can be better accommodated into the major groove of PT-DNA to recognize the $G_{PS}GCC$ core sequence (Fig. 3d).

**The sulfur atom in PT-DNA rotates ~80° upon SBD binding.** ScoMcrA binding also leads to a significant conformational change of PT-DNA. In comparison with the DNA strand that is not in contact with ScoMcrA-SBD, the sulfur atom in the SBD-bound strand of PT-DNA is rotated about the phosphodiester backbone by 80° outward so that it fits better into the ScoMcrA-SBD sulfur-binding cavity (Fig. 4a). This is reminiscent of the flipping of the 5-methylcytosine in methylated DNA to the outside of the double helix by SRA domains[18–21]. Moreover, comparison of our SBD-bound PT-DNA structure with the recently reported free $R_p$ PT-DNA structure[39] also shows that the ScoMcrA-SBD-bound sulfur atom in PT-DNA exhibits an 83° rotation outward upon its recognition by ScoMcrA-SBD (Fig. 4b).

**ScoMcrA-SBD homologs are widely spread in prokaryotes.** Because ScoMcrA is the only identified protein so far that recognizes PT-DNA, we speculated whether other ScoMcrA-like PT-DNA readers might exist. Intriguingly, an extensive position-specific iterated BLAST (PSI-BLAST) search using the sequence of ScoMcrA-SBD revealed that many proteins possessing SBD-homologous domains are widely present among prokaryotes. As many as 1059 sequenced species from 14 phyla of bacteria were found to possess SBD domains, including the more common phyla, such as proteobacteria, actinobacteria, bacteroidetes, firmicutes, and cyanobacteria, as well as less familiar ones including chloroflexi, planctomycetes, verrucomicrobia, acidobacteria, nitrospirae, deinococcus-thermus, rhodothermaeota, balneolaeota, and lentisphaerae (Fig. 5 and Supplementary Table 2). In some well-known orders such as streptomycetales, enterobacterales, and vibrionales, ScoMcrA-SBD homologs are found in even more than 25% of sequenced species (Fig. 5).

Sequence alignment of these putative SBD domains shows that some of the sulfur-recognizing residues of ScoMcrA-SBD, such as P165 and H116 are the most highly conserved (Fig. 6a, Supplementary Fig. 15a and Supplementary Table 3). R117, the positively charged residue forming electrostatic interaction with the sulfur atom in PT-DNA, is replaced by a similarly positively charged lysine in most of the SBD homologs (Supplementary Fig. 15a). Presumably, these SBD homologs use lysines at this position to form electrostatic interaction with the PT-DNA sulfur.

In support of this observation, point mutation of R117K did not appreciably affect the binding affinity between ScoMcrA-SBD and PT-DNA, as measured in fluorescence polarization assays (Supplementary Fig. 15b). On the other hand, the R117Q mutation decreased the association between ScoMcrA-SBD and PT-DNA presumably because glutamine is not as positively charged as arginine and lysine (Supplementary Fig. 15b).

Among the residues in the sulfur-binding pocket, Y164 is unique in that it plays two roles: employing its β-methylene group to form van der Waals interaction with the sulfur atom, and using its hydroxyl group to make hydrogen bonds with the N7 atom of $G^{5'}$ and the N4 atom of $C^5$. PT-DNA sequences in other bacteria possess consensus motifs other than $G_{PS}GCC/G_{PS}GCC$, therefore it is not surprising to find that Y164 is only conserved in a subset of SBD homologs (7 out of the 23 SBD homologs in Supplementary Fig. 15a). Most likely, the SBD homologs in bacteria containing $G_{PS}AAC/G_{PS}TTC$, $G_{PS}ATC/G_{PS}ATC$, and $C_{PS}CA$ core sequences employ residues different from tyrosine at this position to recognize these consensus PT-DNA sequences. Interestingly, the Y164M mutation strengthened the binding between ScoMcrA-SBD and PT-DNA (Supplementary Fig. 15b), presumably because the hydrophobicity of methionine is not less than that of tyrosine, and methionine also possesses substantial conformational flexibility.

A168 contributes to the association with PT-DNA by using its β-methyl group to make a hydrophobic interaction with the sulfur atom of PT-DNA. Other amino acids can replace alanine at this position, since they possess β-methylene groups which can function in the same way. Therefore, it is not surprising that A168 is not as conserved as other sulfur-binding residues (Supplementary Fig. 15a), and mutations of A168 to other residues, such as A168H or A168R in ScoMcrA-SBD did not substantially decrease its binding affinity for PT-DNA (Supplementary Fig. 15b).

Besides these residues, there are other highly conserved residues of ScoMcrA-SBD, including P118, V119, L120, L121, F166, W167, L169, and W175 (Fig. 6a and Supplementary Fig. 15a). These residues do not contribute to the interaction between ScoMcrA-SBD and PT-DNA, but participate in forming the hydrophobic core of the ScoMcrA-SBD domain (Supplementary Fig. 16). Presumably, mutations of these residues might compromise the folding of the ScoMcrA-SBD domain, which could explain their high conservation among homologs.

In contrast, residues of ScoMcrA-SBD-mediating base-specific $G_{PS}GCC$ core sequence contacts, such as S187, R190, and R191, exhibit a higher level of sequence variance. Mapping of the sequence conservation onto the surface of the ScoMcrA-SBD structure reveals that the sulfur-binding cavity is the most

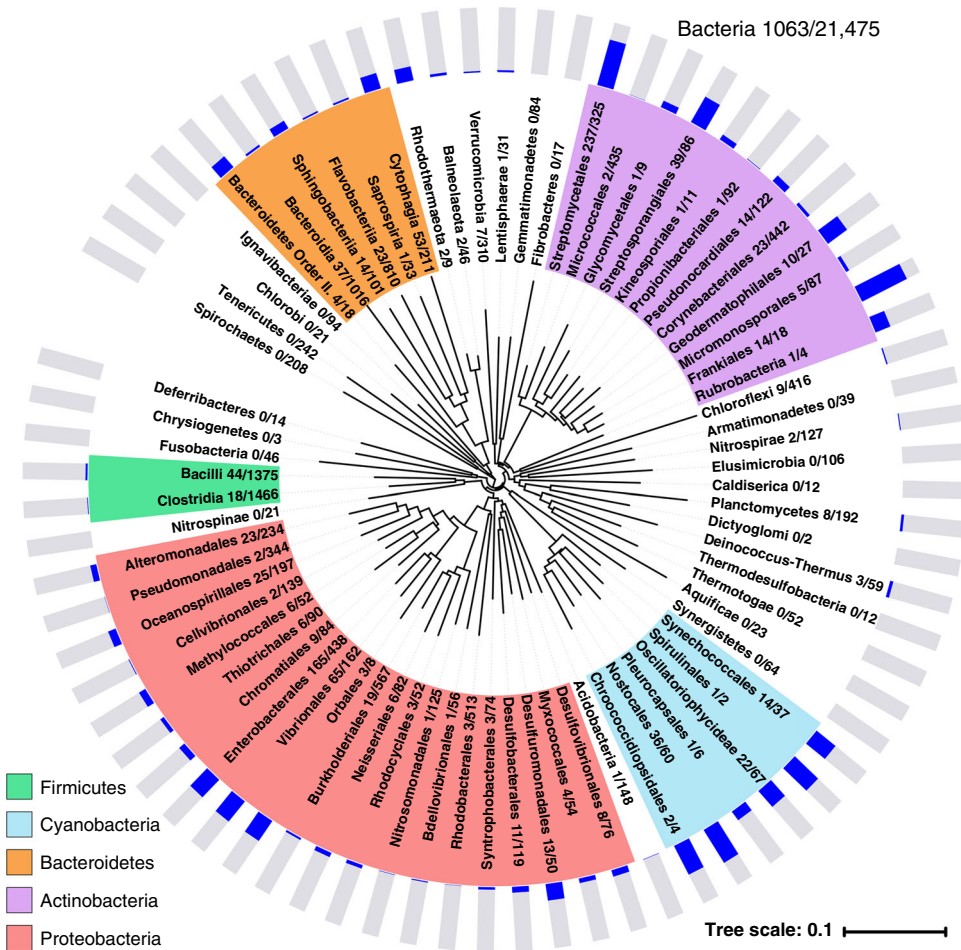

**Fig. 5** Phylogenetic tree for prokaryotic ScoMcrA-SBD homologs. The two numbers behind each phylum/class/order represent the number of species possessing ScoMcrA-SBD homologs and that whose genomes have been sequenced. The occurrence frequency of ScoMcrA-SBD homologs, which was calculated by dividing the number of species possessing ScoMcrA-SBD homologs by the number of sequenced species, for different phyla/classes/orders in the bacteria kingdom was indicated. Details please see Supplementary Table 2

conserved region, which also highlights its functional importance. Noticeably, other parts of the ScoMcrA-SBD surface do not display a comparable level of conservation (Fig. 6b). Therefore, the ScoMcrA-SBD homologs presumably also employ a sulfur-binding cavity containing P165/H116/R117/Y164-equivalent residues to recognize the $R_P$ sulfur atom in PT-DNA.

**SBD homologs bind PT-DNA and restrict gene transfer.** The sequences of the ScoMcrA-SBD homologs found by the BLAST search are highly similar to that of ScoMcrA-SBD, especially for the sulfur-recognizing residues P165, Y164, H116, and R117 (Fig. 6a, Supplementary Fig. 15a, and Supplementary Table 3). To investigate whether these ScoMcrA-SBD homologs could indeed bind PT-DNA, we selected three SBD homologs from *Streptomyces gancidicus*, *E. coli*, and *Morganella morganii*. ScoMcrA originates from *S. coelicolor*, which belongs to the phylum of actinobacteria. On the other hand, both *E. coli* and *M. morganii* belong to the phylum of proteobacteria, which is a separate phylum different from the actinobacteria phylum. Moreover, the *E. coli* and *M. morganii* homologs are distantly related to ScoMcrA (Supplementary Fig. 17). These three ScoMcrA-SBD homologs were heterologously expressed in the *E. coli* strain BL21 (DE3), purified, and were examined for their abilities to interact with PT-DNA with EMSA and fluorescence polarization assays in vitro. Indeed, the purified proteins of these SBD homologs bound PT-DNA with the sulfur atom in the $R_P$ but not $S_P$

configuration in sequence contexts of $G_{PS}GCC/G_{PS}GCC$, $G_{PS}AAC/G_{PS}TTC$, or $G_{PS}ATC/G_{PS}ATC$. Interestingly, the *S. gancidicus* SBD homolog produced only one shifted band with PT-DNA on the gel, whereas the *E. coli* and the *M. morganii* SBD homologs formed two shifted bands with PT-DNA. This indicates that they might interact with palindromic PT-DNA to yield one-protein-plus-one-DNA heterodimeric or two-proteins-plus-one-DNA heterotrimeric complexes (Fig. 6c, Supplementary Fig. 28). In addition, the *S. gancidicus* and the *E. coli* SBD homologs associated equally with $G_{PS}GCC/G_{PS}GCC$, $G_{PS}AAC/G_{PS}TTC$, and $G_{PS}ATC/G_{PS}ATC$, whereas the *M. morganii* SBD homolog interacted more strongly with $G_{PS}GCC/G_{PS}GCC$ (Fig. 5d), suggesting that different SBD homologs possess diverse PT-DNA-binding specificities. Mutation of the critical proline residues corresponding to ScoMcrA-P165 abolished or significantly diminished the association of the SBD homologs from *E. coli* and *S. gancidicus* with PT-DNA, as shown with the fluorescence polarization assays (Fig. 6d).

As a further functional in vivo assay, we examined the abilities of the heterologously expressed SBD homologs to restrict transfer of the *dnd* gene cluster from *Salmonella enterica* serovar Cerro 87. This contains the writer genes of $G_{PS}AAC/G_{PS}TTC$ phosphorothioate modification in DNA, *dndA* through *dndE*[40–43]. The transformation efficiency of the *dnd* gene cluster to a host expressing the SBD homolog from *S. gancidicus* or *E. coli* significantly dropped by 1000 fold compared to a control vector,

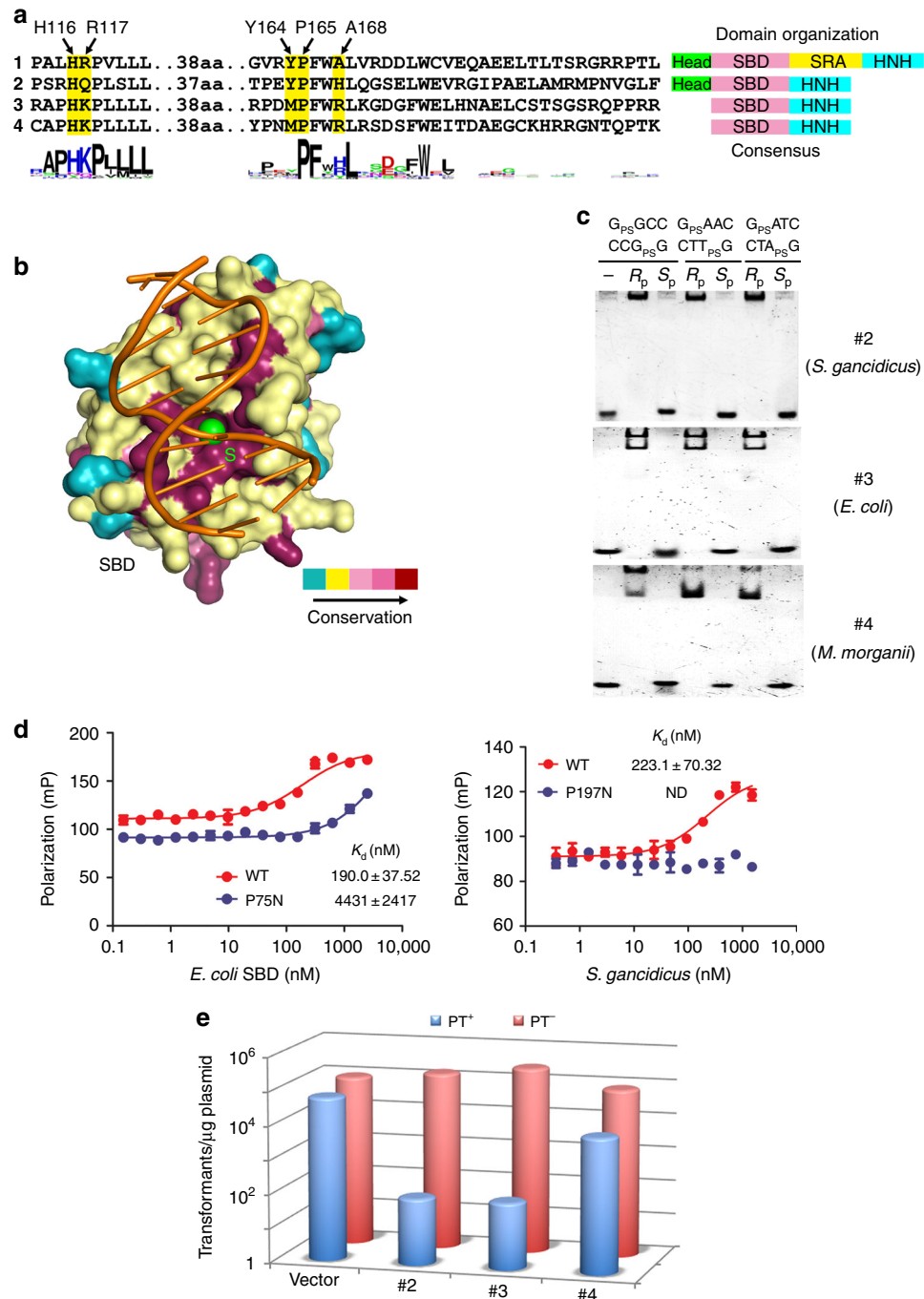

**Fig. 6** SBD-homologous domains might function to recognize PT-DNA. **a** Multiple sequence alignment of representative ScoMcrA-SBD homologs from *Streptomyces coelicolor* (#1), *Streptomyces gancidicus* (#2), *Escherichia coli* (#3), and *Morganella morganii* (#4). The sulfur-recognizing residues are highlighted in yellow. The consensus sequence is shown at the bottom. Right: Domain organizations of these ScoMcrA homologs. **b** The surface of ScoMcrA-SBD is colored according to conservation scores, which shows that its sulfur-recognizing cavity is highly conserved. **c** Purified ScoMcrA-SBD domain homologs from *Streptomyces gancidicus* (#2), *Escherichia coli* (#3), and *Morganella morganii* (#4) specifically recognized PT-DNA with the sulfur atom in the $R_p$, but not in the $S_p$, configuration. **d** Mutation of P482N in the *Escherichia coli* SBD homolog significantly diminished, while mutation of P607N in the *Streptomyces gancidicus* SBD homolog disrupted, the association with PT-DNA with the core sequence $G_{PS}AAC$. **e** Heterologous expression of *scomcrA* homologs from *S. gancidicus* (#2), *E. coli* (#3), and *M. morganii* (#4) restricted transfer of the *dnd* gene cluster from *Salmonella enterica*, which contains the genes encoding the "writer" proteins of DNA phosphorothioation. Transformation frequencies of empty pBluescript vector (PT$^-$) and that harboring the *dnd* gene cluster (PT$^+$) into *E. coli* DH10B expressing various *scomcrA* homologs are shown

while a 10-fold decrease in transformation efficiency of the *dnd* gene cluster into a host harboring the *M. morganii* SBD homolog was observed (Fig. 6e). Therefore, our in vivo and in vitro studies support the hypothesis that these prokaryotic ScoMcrA-SBD homologs function to specifically recognize PT-DNA.

## Discussion

Sulfur is a vital constituting element for many building blocks of life, such as amino acids, tRNA, coenzymes/cofactors, etc. However, to our knowledge, few crystal structures are available that describe the direct recognition mechanism for sulfur atoms in

various sulfur-containing molecules by their binding proteins. Our study reveals that in addition to an irregular loop that binds in the major groove of $G_{PS}GCC$, a highly conserved cavity is employed in the SBD domain to recognize the phosphorothioate in PT-DNA, with both hydrophobic and electrostatic interactions. Similar to other DNA–protein complexes, electrostatic interactions are formed by the positively charged guanidinium group of R117, in particular with the non-bridging oxygen atom of phosphorothioate in PT-DNA since the electronegativity of oxygen is higher than that of sulfur. In addition, the sulfur atom, which is low in electronegativity, is compensatorily surrounded by a relatively hydrophobic environment including the pyrolidine ring of P165, the β-methylene groups of Y164 and H116, and the β-methyl group of A168. It should also be pointed out that in terms of entropy the sulfur atom of PT-DNA would be more favorable to be enclosed by the hydrophobic cavity of ScoMcrA-SBD than to be exposed to solvent. According to quantum calculation at the MN15/6-31 + G* level of density functional theory (DFT), the overall interaction energy for $R_P$-phosphorothioate was estimated to be stronger by about 16 kcal/mol than that of its $S_P$-chiral cousin in such an environment, almost equivalent to that of a phosphate in normal DNA (Supplementary Fig. 18). Moreover, the solvation energy for either $R_P$-phosphorothioate or $S_P$-phosphorothioate was poorer by 9–11 kcal/mol than that of a normal phosphate in aqueous solution, at the same level of DFT calculation with Truhlar and coworkers' SMD solvation model[44]. This recognition pattern was discovered for PT-DNA and SBD. It is expected that similar recognition mechanism might account for many biological functions of the endogenous phosphorothioation in bacterial genomes (also see the comparison and contrast of oxygen and sulfur atoms in Supplementary Discussion).

The electronic structure of phosphorothioate has been well characterized in earlier work[12,29]. In unbound phosphorothioated DNA, the negative charge favors the sulfur atom in an anionic form $[>P(=O)S^-]$, similar to the phosphate group in normal DNA (see Fig. 6 in ref. [12]). On the other hand, the amount of electron density of the sulfur atom is distributed in a much larger region in phosphorothioate group than that of the oxygen atom in phosphate group. Owing to a larger radius and higher polarizability, the sulfur atom favors a larger London dispersion force and less electrostatic interaction. SBD protein provides two interacting sites, one from R117 (positively charged, salt-bridge hard partner) and another from hydrophobic cavity (uncharged, van der Waals interaction dominantly, soft partner). Phosphorothioate binds R117 with the hard P–O side and hydrophobic cavity with the soft P–S side. Therefore, we suggest that the sulfur atom of PT-DNA is bound in a hydrophobic uncharged cavity by dispersion force.

ScoMcrA binds to and cleaves 5mC-DNA modified by Dcm[18–27]. A domain sandwiched between the SBD and the HNH domain in ScoMcrA structure has an overall structure similar to classical SRA, but does not possess the NKR finger and thumb features[18–20]. Besides, the residues recognizing 5-methylcytosine in UHRF1/SUVH5 are not conserved in ScoMcrA-SRA. In this term, ScoMcrA-SRA may interact with 5mC-DNA to achieve discrimination of DNA methylation from unmodified DNA in a way different from the known SRA domains (Supplementary Fig. 19). In comparison with eukaryotic SRA, the structure of ScoMcrA-SRA is more similar to prokaryotic ones. The root-mean-square-deviation value between ScoMcrA-SRA and MspJI-SRA[22] is 9.228 Å, lower than that of 12.032 Å between ScoMcrA-SRA and UHRF1-SRA[20].

Sequence comparisons of the ScoMcrA homologs showed that most of them possess putative HNH endonuclease domains (Fig. 6a and Supplementary Fig. 15a). In particular, the active site residues of H508/N522/H531, characteristic of the HNH motif,

and the zinc finger fold composed of the residues of C484/C489/C527/C530 are highly conserved (Supplementary Fig. 20). In ScoMcrA, SBD, SRA, and HNH have a linear spatial organization in three-dimensional space, with SBD farther than SRA from HNH (Supplementary Fig. 21), which accounts for the observation that ScoMcrA cleaves DNA at a site further away from the phosphorothioate linkage than from the methylation site[14].

DNA modifications are generally recognized by small to medium-sized modular domains, such as SRA, which are usually linked with domains that either perform specific function or recruit other proteins to translate the information encoded by DNA modifications. In support of this notion, our work provides structural evidence that two distinct DNA modification recognition domains in one protein share a HNH endonuclease domain to perform DNA cleavage. Such domain organization hints that it is possible to fuse two or more different DNA modification-recognition domains in a tandem way with an endonuclease catalytic domain, i.e., HNH, in our case. The engineered hybrid enzyme has potential application in large-scale industrial fermentation to prevent infection of phages with varied DNA modifications.

The SBD homolog from *S. gancidicus* displayed flexibility in the selection of substrate PT-DNA with different core sequences (Supplementary Fig. 22). This feature can be utilized to enrich PT-DNA from diverse environment to survey the relationship between the distribution or abundance of PT-DNA and environmental conditions. The fact that SBD homologs are present in at least 14 phyla of bacteria and three of them were indeed verified to associate with PT-DNA in vitro suggest that the SBD domain might function as a widely existing specific PT-DNA reader in prokaryotes. As an interesting hypothesis, potential information encoded by the PT modification might be deciphered through its specific interaction with the SBD domain and thereby transferred to other SBD-conjugated domains, such as HNH, to achieve its biological functions.

Cleavage by ScoMcrA occurs at ~23 bp away from the phosphorothioate linkage at the 5" side of $G_{PS}GCC$ (Supplementary Fig. 23). By combining the structures of FL ScoMcrA and SBD-SRA in complex with PT-DNA, we obtained a model of FL ScoMcrA dimer recognition and cleavage of PT-DNA (Supplementary Figs. 24 and 25), which agrees well with our cleavage data. Our model suggests two ScoMcrA might not bind to two strands of one DNA at the same time (Supplementary Fig. 26).

DNA phosphorothioation is a fascinating phenomenon that holds high promise for potential application in gene silencing[45] and genome editing[46,47]. Phosphorothioated antisense oligonucleotides have also recently been used to target human telomerase for cancer therapy[48]. Our work might suggest clues for designing new generations of Cas9-like genome editing nucleases through utilizing the special PT-DNA recognition expertize of SBD domains.

## Methods

**Molecular cloning.** Strains, plasmids, primers, and oligonucleotides used are listed in Supplementary Table 4. *E. coli* DH10B (Thermo Fisher) and *Escherichiacoli* BL21(DE3) (Novagen) was grown in Luria Broth medium supplemented with 100 mg/mL ampicillin or 34 mg/mL chloramphenicol as required. Genes encoding *S. coelicolor* FL ScoMcrA, the SBD-SRA domain of ScoMcrA (residues 91–442), and the SBD domain of ScoMcrA (residues 91–260) were amplified by high-fidelity PCR with KOD-Plus DNA polymerase (Toyobo) using primer pairs ScoFL-F/ScoFL-R, SBDSRA-F/SBDSRA-R, and SBD-F/SBD-R, respectively. The PCR products were cleaved by BglII/EcoRI or BamHI/EcoRI (New England Biolab) and inserted into the pSJ8 plasmid using T4 DNA ligase (Takara), resulting in pJTU1660, pJTU1668, and pJTU1669.

The 1380 bp DNA fragment encoding the *E. coli* ScoMcrA homolog (WP_000199891) along with its promoter was synthesized by GeneScript according to the sequence in NCBI GenBank (4690337–4691716 of CP006584.1). The genes encoding the *M. morganii* ScoMcrA homolog (WP_032098169) and the *S.*

*gancidicus* ScoMcrA homolog (WP_006134840) were synthesized according to their protein sequences with codons optimized for *E. coli* (Supplementary Table 5).

Genes encoding the SBD domain of the *E. coli* ScoMcrA homolog (residues 1–170) and the SBD domain of the *M. morganii* ScoMcrA homolog (residues 1–170) were amplified by high-fidelity PCR using primers Eco-SBD-F/Eco-SBD-R and Mmo-SBD-F/Mmo-SBD-R and cloned into pSJ8 as above, resulting in pJTU1670 and pJTU1671, respectively. The gene encoding the FL *S. gancidicus* homolog was amplified by high-fidelity PCR using primers Sga-F/Sga-R and cloned into pSJ8 as above, resulting in pJTU1672.

Site-directed mutagenesis of pJTU1669 was performed by using a Hieff Mut™ site-directed mutagenesis kit (YEASEN) with primers H116I-F/H116I-R, R117A-F/R117A-R, R117G-F/R117G-R, Y164I-F/Y164I-R, P165N-F/P165N-R, A168G-F/A168G-R, A168I-F/A168I-R, Y164F-F/Y164F-R, S187A-F/S187A-R, and R191A-F/R191A-R, resulting in pSBD-H116I, pSBD-R117A, pSBD-R117G, pSBD-Y164I, pSBD-P165N, pSBD-A168G, pSBD-A168I, pSBD-Y164F, pSBD-S187A, and pSBD-R191A, respectively. All mutants were confirmed by DNA sequencing.

**Protein expression and purification**. The pSJ8 derivatives were transformed into *E. coli* BL21(DE3). For protein overexpression, 10 mL of the overnight culture was inoculated into 1 L LB medium supplied with 100 mg/mL ampicillin. Next, the culture was incubated at 37 °C to $OD_{600} = 0.4$ and induced by 0.2 mM IPTG for 12 h at 16 °C. Then, the cells were centrifuged, resuspended in $Ni^{2+}$ column-binding buffer (20 mM Tris–HCl, pH 8.0, 300 mM NaCl, 20 mM imidazole), and lysed by a cell homogenizer (JNBio) at 4 °C. After centrifugation (16,000 × *g* for 30 min at 4 °C), the supernatant was applied to a HisTrap HP column (GE Healthcare) and purified using an AKTA FPLC (GE Healthcare) by eluting with imidazole linear gradient 20–500 mM. The MBP-tagged protein products were cleaved by TEV protease, followed by purification with a HiTrap Heparin HP affinity chromatography column, a Source 15Q anion exchange chromatography column, and a Superdex 200 GL 10/300 gel filtration chromatography column. The product of pJTU1670 resulted in precipitation after TEV protease cleavage, so the MBP-tagged protein was used instead. The gel filtration buffer contained 10 mM Tris–HCl, pH 8.0, 100 mM NaCl, and 2 mM dithiothreitol (DTT). Peak fractions were combined and concentrated to 10 mg/mL. Purified proteins were visualized by Coomassie-stained 12% SDS–PAGE analysis, and protein concentration was determined using a Bradford Protein Assay Kit (Bio-Rad).

**Crystallization and structure determination**. Crystals of FL *S. coelicolor* ScoMcrA were grown at 14 °C by the hanging-drop vapor-diffusion method with drops consisted of 1 μL protein (10 mg/mL) and 1 μL reservoir solution. The reservoir solution contained 0.2 M lithium sulfate, 0.1 M Tris–HCl, pH 8.5, and 13% (w/v) polyethylene glycol (PEG) 4000. Crystals were transferred to the crystallization buffer supplemented with 25% glycerol before being flash-frozen. An X-ray diffraction data set at the 3.15 Å resolution was collected at the beamline BL17U1 at Shanghai Synchrotron Radiation Facility (SSRF, China), using an ADSC Quantum 315r CCD area detector[49]. The diffraction data were processed using the HKL2000 software[50]. The crystal belonged to the space group $P2_12_12_1$, and contained six molecules of ScoMcrA in each asymmetric unit. The crystal structure was determined by the SAD method with the PHENIX program[51], using a SeMet derivative of FL ScoMcrA. After model-building by COOT[52] and refinement by the CCP4 program REFMAC5[53,54], the final model includes residues 5–59, 80–107, and 114–560 for chain A; residues 4–29, 33–60, 78–107, 114–427, and 432–560 for chain B; residues 7–29, 40–59, 78–107, 114–430, and 432–560 for chain C; residues 9–26, 38–59, 77–107, 114–175, and 206–560 for chain D; residues 10–20, 36–60, 76–107, 114–180, 192–201, 205–428, and 431–560 for chain E; residues 94–108, 111–429, and 433–560 for chain F.

Crystals of the SBD-SRA domain of ScoMcrA (residues 91–442) in complex with the $R_p$ form of 10 bps double-stranded palindromic PT-DNA oligonucleotide (annealed by the $R_p$ forms of oligonucleotide GGCC10, 5′-C-C-C-G-($R_p$ PS)-G-C-C-G-G-G-3′) were grown at 14 °C by the hanging-drop vapor-diffusion method with drops consisted of 1 μL sample (10 mg/mL protein and 0.8 mg/mL DNA) and 1 μL reservoir solution. The reservoir solution contained 2.0 M ammonium sulfate, 0.1 M CAPS buffer, pH 10.5, and 0.2 M lithium sulfate. Crystals were cryoprotected as above, and a diffraction dataset at the resolution of 3.30 Å was collected and processed at the BL17U1 beamline at SSRF, China[49]. The crystal belonged to the space group $I4_122$, and contained one molecule of ScoMcrA-SBD-SRA in complex with one molecule of PT-DNA in each asymmetric unit (ScoMcrA-SBD-SRA actually forms a dimer in the crystal with the two protomers related by the crystallographic two-fold rotation axis). The crystal structure was determined by the molecular replacement method with the Phaser program[53,55] using the structures of the SBD domain and the SRA domain in FL ScoMcrA by itself as the searching model. After refinement by the CCP4 program REFMAC5[53,54], the final model includes residues 91–418 of ScoMcrA and all the nt in PT-DNA.

Crystals of the SBD domain of ScoMcrA (residues 91–260) in complex with the $R_p$ form of 8 bp double-stranded palindromic PT-DNA (annealed by the $R_p$ form of oligonucleotide GGCC8, 5′-C-C-C-G-($R_p$ PS)-G-C-C-G-G-3′) were grown at 14 °C by the hanging-drop vapor-diffusion method with drops consisted of 1 μL sample (10 mg/mL protein and 2.8 mg/mL DNA) and 1 μL reservoir solution. The reservoir solution contained 0.1 M HEPES, pH 7.5, 0.8 M ammonium phosphate monobasic, 0.8 M potassium phosphate monobasic, and 1% glycerol. Crystals were

cryoprotected as above, and a diffraction dataset at the resolution of 1.70 Å was collected at the BL19U1 beamline at National Center for Protein Science Shanghai (NCPSS), China. The crystal belonged to the space group $C2_1$, and contained one molecule of ScoMcrA-SBD in complex with one molecule of PT-DNA in each asymmetric unit. The crystal structure was determined by the molecular replacement method with the Phaser program[53,55] using the structure of the SBD domain in FL ScoMcrA by itself as the searching model. After refinement by the CCP4 program REFMAC5[53,54], the final model includes residues 94–251 of ScoMcrA and all the nt in PT-DNA. Quality of the models was checked with the CCP4 program Procheck[53].

**Preparation of stereo-specific PT-DNA**. The PT-DNA oligonucleotides were chemically synthesized and PAGE-purified by Sangon Biotech Co. Ltd. (Shanghai, China). The $R_p$ and $S_p$ stereoisomers were separated by anion exchange HPLC with a DNAPac PA-100 analytical column (Thermo-Fisher Scientific) on an Agilent 1260 infinity series system at a flow rate of 1 mL/min with the following parameters (column temperature: 50 °C; solvent A: 10 mM Tris–HCl, pH 8.0; solvent B: 10 mM Tris–HCl, pH 8.0, 1 M NaCl; gradient: 5% B for 5 min, 20% B to 50% B over 30 min; 100% B for 10 min; detection: UV absorbance at 260 nm). The eluant was desalted with a Copure C18 column (Biocomma), dried on an RVC 2-25 rotational vacuum concentrator (Christ), and dissolved with distilled de-ionized water. The concentration of oligonucleotides was determined by spectrophotometric measurement on a NanoDrop 2000 spectrophotometer (Thermo-Fisher Scientific), and double-stranded DNA was prepared by mixing equal molar concentrations of complementary oligonucleotides, followed by heating to 95 °C for 2 min and gradual cooling.

**Electrophoretic mobility shift assay**. One EMSA reaction contained 20 ng DNA (50 ng for the GGCC substrates with SBD and its point mutants) and protein with concentration two-fold or four-fold that of DNA (molar ratio) in a buffer composed of 20 mM Tris–HCl, pH 8.5, 200 mM NaCl, 1 mM DTT, and 5% glycerol in a total volume of 20 μL. After incubation at 4 °C for 30 min, the reaction mixtures were loaded onto 12% non-denaturing polyacrylamide gels (with the ratio of acrylamide:bisacrylamide being 79:1, w/w) and electrophoresed on a BG-verMINI electrophoresis cell (Baygene) in 0.5 × TBE buffer at 150 V for 90 min. Gels were stained by SYBR Gold (Invitrogen) and imaged on a Gel Doc XR+ Molecular Imager (Bio-Rad).

**Transformation efficiency assay**. The DNA fragment encoding FL *E. coli* ScoMcrA homolog with its promoter was amplified by high-fidelity PCR using primers EcoHE-F/EcoHE-R, and the fragment was cloned into the EcoRV site of pACYCDuet™-1 by blunt end ligation, resulting in pJTU1673. The DNA fragments encoding *M. morganii* ScoMcrA homolog or *S. gancidicus* ScoMcrA homolog with *E. coli* promoter were obtained through splicing by overlap extension (SOE) PCR with 504 bp fragment containing the promoter of *E. coli* ScoMcrA homolog (amplified using primers EcoHE-F/P-Mmo-R or EcoHE-F/P-Sga-R), and FL *M. morganii* ScoMcrA homolog or FL *S. gancidicus* ScoMcrA homolog fragment (amplified using MmoHE-F/MmoHE-R or SgaHE-F/SgaHE-R). The fragments were cloned into pACYCDuet™-1 as above, resulting in pJTU1674 and pJTU1675.

The pACYCDuet™-1 vector and its derivatives carrying FL *E. coli*, *M. morganii*, and *S. gancidicus* ScoMcrA homologs with *E. coli* promoters (pJTU1673, pJTU1674, and pJTU1675) were introduced to *E. coli* DH10B, and competent cells of the resulting strains were prepared using the standard calcium chloride protocol. Transformation frequency was determined by introducing 100 ng pBluescript SK+ ($PT^-$) or pJTU1238 ($PT^+$) plasmid DNA, which carries the *dnd* gene cluster from *Salmonella enterica* serovar Cerro 87[56], to the competent cells. The number of *E. coli* colony forming units in each experiment was determined by making serial dilutions and plating LB agar plates. Each experiment was repeated three times and the mean value of the transformation frequency was reported.

**Molecular graphics**. All protein structure figures were generated by the PyMOL program (http://www.pymol.org). Sequence conservation of ScoMcrA mapped onto the surface of its crystal structure was generated by the ConSurf server (http://consurf.tau.ac.il)[57].

**Construction of the phylogenetic tree**. The phylogenetic tree in Fig. 5 was constructed using MEGA7 software based on the 16S rRNA-encoding DNA sequences of representative organisms in all the known phyla of bacteria kingdom. The tree for SBD-containing homologs (Supplementary Figure 17) was constructed by Clustal Omega (https://www.ebi.ac.uk/Tools/msa/clustalo/) based on the homology of the amino acid sequences. All the parameters for tree construction are the default values by the online softwares. Sequences were compared by MUSCLE or Clustal Omega, and the neighbor-joining method was employed.

We used the protein sequence of ScoMcrA−SBD (residues 91–260) to perform the PSI-BLAST search in the National Center for Biotechnology Information (NCBI) database, with the threshold set as the default value of 0.005. As a result, 2761 proteins containing sequences homologous to ScoMcrA-SBD were obtained, and these belonged to 1063 bacteria species; for example, many SBD homologs could be found in different *E. coli* strains, but they all belonged to the same species,

*E. coli*. After removing those annotated as "unclassified bacteria", 1059 species remained that contained SBD homologs. As of April 10, 2018, there were 21,745 species in the bacteria kingdom whose genomes had been sequenced, according to the NCBI taxonomy database. Therefore, in Fig. 5, "Bacteria 1059/21,745" means that among the 21,745 bacteria species whose genomes have been sequenced, 1059 (4.9% of 21,745) species possess SBD homologs. The same statistical method was used to search for SBD homologs in each phylum, class, and order in the bacteria kingdom, and the statistics of number of species possessing SBD homologs/number of sequenced species was shown in the phylogenetic tree. For example, "Streptomycetales 237/325" means that in the order of Streptomycetales, 325 species have their genomes sequenced and among them 237 species possess SBD homologs. The occurrence frequency of SBD homologs (which is calculated through dividing the number of species possessing SBD homologs by the number of sequenced species) was calculated.

**Fluorescence polarization assay**. The binding of DNA and protein were measured using a fluorescence polarization assay. 5 nM 5-carboxyfluorescein-labeled double-stranded DNA probe with various concentrations of SBD protein was incubated in buffer containing 20 mM Tris (pH 8.0), 50 mM NaCl, 1 mM DTT, 5% glycerol at room temperature. We used a SpectraMax i3x to measure the fluorescence polarization. The dissociation constants ($K_d$) were derived from the equation $mP = [\text{maximum } mP] \times [C]/(K_d + [C]) + [\text{baseline } mP]$, where mP is millipolarization, and [C] is protein concentration. Curves were performed individually using the equation, the curves and $K_d$ values were analyzed by using Graphpad prim software (version 5.0). Our data were obtained from two or four experimental replicates.

**Calculation of interactions between PT-DNA and SBD**. The inter-molecular interaction for PT-DNA and SBD protein was computed using a variety of different quantum chemical methods and a model system with phosphorothioate and neighboring residues from α-helix 5 (residues 114–117, ALHR) and α-helix 8 (residues 164–168, YPFWA). For simplification, the side chains of F, W, and L were trimmed owing to no direct interaction with phosphorothioate, but a water molecule was retained by its special hydrogen bonding network within α-helix 5. The model system was partially optimized with the ONIOM method at the level of B3LYP/6-31 + G*|PM6[58], in which phosphorothioate and guanidinium groups were set up as the core higher-level quantum calculations with the bridging water molecule and contacted peptide bond of P165 for the electrostatic and van der Waals interactions. The $R_p$-phosphorothioate complex model was well-maintained after the above optimization, and then the $S_p$-phosphorothioate and normal phosphate models were constructed by replacement of oxygen and sulfur atoms. The resulting three models were shown in Supplementary Fig. 17.

The interaction energy for the complexes was computed at the B3LYP/6-31 + G* level of theory, with the BSSE correction[59]. The overall interaction in the gas phase was estimated to be −131.8, −117.1, and −135.6 kcal/mol, and −121.5, −107.3, and −125.3 kcal/mol before and after the counterpoise correction, in the cases of $R_p$-phosphorothioate, $S_p$-phosphorothioate, and normal phosphate complexes, respectively. The primary contribution belonged to the electrostatic interaction between phosphorothioate (phosphate in normal DNA) and R117's guanidinium group. Apparently, the $R_p$-phosphorothioate interacted with the SBD protein much better than its $S_p$-configuration cousin, and very close to the normal phosphate in the interaction energy. This is in good agreement with the well-known fact that sulfur is lower in electronegativity than oxygen, and the $R_p$-complex preserved most of the electrostatic contact of P–O and guanidinium as in the normal phosphate model. On the other hand, oxygen–sulfur switch would lead to a significant loss of energy, 14 kcal/mol, in the case of the $S_p$-phosphorothioate complex.

The solvation energy was calculated at the same DFT level of theory, with the SMD solvation model[60]. It was found that poor solvation for phosphorothioate was the second important factor for the molecular recognition. The solvation energy was calculated to be −81.1, −79.1, and −89.8 kcal/mol for the three cases of $R_p$-phosphorothioate, $S_p$-phosphorothioate, and normal phosphate complexes, respectively. So the solvation energy decreased by 9–11 kcal/mol after phosphorothioation. This is also consistent with the lower electronegativity of sulfur, and thereby poorer polarization in aqueous solution. Upon the complex formation, sulfur was fully buried inside of SBD protein and the complex was expected to be very similar from phosphorothioate to phosphate in solvation energy.

In short, the molecular recognition between PT-DNA and ScoMcrA-SBD stems from two factors, a good orientation for salt-bridge and a poor solvation of isolated phosphorothioate. The SBD hydrophobic cavity leverages the solvent effect and enhances the molecular interaction between PT-DNA and ScoMcrA-SBD.

## Data availability

The coordinates and structure factor files of full-length *Streptomyces coelicolor* ScoMcrA by itself, the SBD-SRA domains of ScoMcrA in complex with PT-DNA, and the SBD domain of ScoMcrA in complex with PT-DNA have been deposited in PDB with accession numbers 5ZMM, 5ZMN, and 5ZMO, respectively. Other data are available from the corresponding authors upon reasonable request.

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

## Acknowledgements

Dr. Neil Price (NCAUR-USDA, Peoria, IL) undertook an initial review prior to publication. We thank Jianhua He, Wenming Qin, Lijie Wu, and other staff at the beamline BL17U1 at Shanghai Synchrotron Radiation Facility (SSRF) and the beamline BL19U1 at National Center for Protein Science Shanghai (NCPSS). This work was supported by grants from the National Natural Science Foundation of China (grant numbers 31670034, 31470223, 31670106, 31130068, 31770070, and 21661140002), and the Program for Professor of Special Appointment (Eastern Scholar) at Shanghai Institutions of Higher Learning.

## Author contributions

G.L., G.W., and X.H. designed the research plan and performed the phylogenetic analysis. G.L. performed protein purification, crystallization, data collection, biochemical characterization, EMSA, and microbial genetics experiments. W.F. performed fluorescence polarization experiments. H.Y. provided assistance. Z.Z., Y.H., and G.L. performed structure determination. X.W. and Y.Z. performed theoretical calculation. Y. W. performed the site-mutagenesis of R117, Y164, and A168. Z.D. provided materials and reagents. G.L., G.W., and X.H. wrote the manuscript.

## Additional information

**Competing interests:** The authors declare no competing interests.

