## [Peer Review File · Nature Communications]

Reviewers' comments:

Reviewer #1 (Remarks to the Author):

Liu et al. describe structural and biochemical characterization of a restriction enzyme ScoMcrA which is capable of specific cleavage of phosphorothioate DNA. The role and recognition of this DNA modification is a very interesting subject. Its studies can not only provide insights into the molecular recognition of nucleic acids but can also help build tools for nucleic acid processing/editing.

The results are presented clearly and the figures help understand the main points. This work is technically solid. The structures are refined to good parameters and the validation reports show satisfactory geometry of the models. The structure of SBD-DNA complex, which is the most important for the conclusions of the manuscript, is solved at relatively high resolution of 1.7 Å. The structural work is verified by mutagenesis studies. The main finding is that the sulphur atom of the phosphorothioate linkage is recognized by interactions with a hydrophobic pocket in the SBD domain. The generality of the proposed mechanism is verified for several SBD domains from various bacterial species. This is also demonstrated by a nice *in vivo* assay.

One element which is missing in the manuscript is the structural analysis of the full length restrictase and in particular its potential interactions with the DNA substrate. A very general model of this interaction is shown in Suppl. Fig. 18., however a more detailed analysis needs to be done.

The biological relevance of the dimeric structure of ScoMcrA alone shown in Fig 1c needs to be discussed. In this dimer, are the SBD residues responsible for phosphorothioate binding, the SRA residues potentially binding methylated DNA and the HNH residues forming the active site accessible? In other words, could they interact with DNA in this dimeric configuration? What conformational changes (if any) would be required for the enzyme to bind and cleave the DNA? How does the arrangement of HNH domains in this dimer relate to the observed cleavage sites (Suppl Fig 20)? In the dimeric structure the two HNH are located close to each other. Does it imply that both will cleave on one side of the phosphorothioate and/or methyl modification? Are there any other dimers that are generated by crystal contacts, the arrangement of which would fit better the cleavage data? How does the arrangement of SRA and SBD domains compare between the structure of full-length ScoMcrA and SBD-SRA fragment with DNA?

The crystallized DNA had two modifications but only one phosphorothioate interacts with the protein. However, *E. coli* and the *M. morgani* SBDs may form dimers on the DNA. Can two SBDs be modelled on one DNA without clashes? Additionally, in order to better understand the mechanism of SDB, the affinities (Kd) of SBD to doubly and singly PT-modified DNA need to be measured.

In Suppl Figure 20a there are many more cuts indicated in panel (a) than bands observed in panel (b). How can this be explained? How was the DNA detected in this gel?

Minor point: Labels in Suppl Fig 2 are swapped.

Reviewer #2 (Remarks to the Author):

The manuscript describes the crystal structure for a protein domain that binds phosphorothioate (PT) DNA. This domain is part of an enzyme, ScoMcrA, that recognizes the PT modification within the context of a conserved DNA sequence motif and cuts the modified DNA. A crystal structure at lower resolution is reported for the whole enzyme, while a high-resolution structure is reported for the domain responsible for binding the PT-modified DNA. This work contributes a structural

understanding of how PT DNA can be recognized.

The writing needs a little polishing to improve the English and make it more readable.

There are a number of inconsistencies that require correction (for example, in the introduction the authors state there are 'three consensus sequence patterns of PT modifications', then proceed to list four such patterns in four different organisms).

There are also major problems with the interpretation presented for the crystal data. For example, the authors state that a positive charge on an arginine residue (R190) forms hydrogen bonds to negatively charged O6 atoms of guanine (G3) and (G4), but also that this positive charge forms a hydrogen bond to the positively charged N4 atom of cytosine (C4). Two positive charges do not form hydrogen bonds to each other – this interpretation of the crystal data is clearly false, as it is simply not physically possible. The authors state that the 'sulfur-recognizing residues' they identify are highly conserved residues within PT-recognizing homologs they identify in sequence databases, yet one of these 'highly conserved' residues occurs only in their enzyme (R117), while the 'gate' residue Y164 is present in only 7 of the 23 homolog enzyme sequences presented in their alignment (Supp. Fig 15). These residues are clearly not highly conserved, as is claimed. It would seem important to at least comment on how the other residues at the Y164 position might or might not function in a similar manner to the proposed role of Y164. In general, the interpretation of the crystal data would benefit from a more precise, accurate analysis, which I hope the authors will undertake and describe more accurately in a revised and improved manuscript.

The topic of how PT modification can be specifically bound is important and interesting, and is well worthy of publication. The presentation and analysis just needs to be tightened up to generate an excellent paper.

Specific comments:

Introduction:

Page 3: State 'three consensus patterns', then list 4 patterns – need to be consistent.

Page 5: states that the phosphorothioate sulfur is negatively charged and forms 'salt bridges' with R117, but then the paper argues this sulfur is bound in a hydrophobic uncharged pocket. How is this consistent (negatively charged and hydrophobic)? This needs an explanation of why a sulfur atom behaves differently than an oxygen atom and how any differences can lead to specific recognition of the sulfur in PT DNA.

Results:

Page 9: would R117 form a similar salt bridge with oxygen rather than sulfur? Is there anything here that would be specific for sulfur over the normal oxygen?

Page 10: Mutation to Proline165: mutating a proline to any other amino acid will change the backbone of the protein, as no other amino acid has the same backbone connectivity as Proline, so best to be careful in interpreting the resulting loss of activity (could be due to improper folding/shape rather than simply charge/no charge).

Page 10: Why not try the R117K mutation that is common in the homologs?

Page 10: Note that A168 is not conserved in other homologs (most have a large, charged amino acid (His or Arg) at this position. The effect of removing the methyl group (A168G) is small. Either homologs have quite different binding pockets, or the role of the residue at this position is not simply a van der Waals interaction, or perhaps is not so important to PT binding as is argued here.

Page 11: Does the reported hydrogen bond of Y164 to the N4 of C5 contribute to opening the sulfur binding pocket and keeping the pocket open? Seems this may be important for PT recognition, and might explain why Y164 is not well conserved in homologs, as these might

recognize PT DNA in a different sequence motif context.

Page 12: The lack of electron density for Y164 when not in complex with PT DNA indicates this residue is flexible and is sampling multiple configurations, not that it is in a closed state – only 2 of 6 molecules have such a closed state.

Page 13: It may be that binding to the GGCC motif causes Y164 to rotate into the 'open' position, not necessarily due to binding to the sulfur of PT DNA as is implied.

Page 13: what axis does the sulfur rotate about?

Page 14: R117 and Y164 are not highly conserved in the alignment shown in Sup Fig 15!
It appears H116, P118, P165, F166, W167, L169, W175 are highly conserved: seems to be too much focus on R117 and Y164, and not enough attention to the other structural elements (P118, F166, W167, L169, W175).

Page 15: How were the three expressed homologs "distant" in relationship to ScoMcrA?

Page 15: If the expressed homologs were distant, why did they recognize all three sequence motif contexts (GGCC, GATC, GAAC)? Do these have an SRGRR loop or equivalent sequence recognition element?

Page 20: HNH endonucleases are efficient – they are not inherently poor at DNA cutting, and PD-ExK endonucleases are not inherently better. This suggestion is unlikely to succeed.

Page 20: There has not been a demonstration of sequence-context independent recognition of PT DNA made, so the suggestion to broadly characterize PT motif using *S. gancidicus* enzyme is premature. If *S. gancidicus* is found to be highly flexible in recognition, what is the basis of that flexibility (compared to ScoMcrA)? Does it lack a DNA recognition loop analogous to SRGRR of ScoMcrA?

Page 20: Agree that PT DNA is widespread and PT-readers are important in processing the modification information.

Figure 2: suggest showing the guanine O6 and N7 and the cytosine N4 positions in the figure (as bumps on the bases) and the contacts to these by the various amino acid residues.

Why are there no contacts to the G6:C6 base pair, which is part of the GGCC specific motif? Seems there should be something specifying recognition for this base pair.

Figure 5: Suggest using residues observed in SBD homolog enzymes to guide mutations made for testing ScoMcrA activity, in addition to mutations expected to K/O the enzyme function (such as R117 to Q or K; Y164 to M; A168 to H or R).

Supp Fig 11 and 12: The DNA does not seem to be base paired in these figures: is this the case? If so, comment on why it is not base paired; otherwise correct the image.

Note: there cannot be a H-bond from positive Arg to a positive N4-cytosine group. The contact pattern here needs to be re-analyzed and corrected.

Supp Fig 13 and 14: Perhaps comment on the observation that Y164 is dynamic, not fixed into a single position (4 of 6 chains in the crystal have flexible Y164, ie no density).

Supp Fig 17: Please make text under the highlights BOLD to make it easier to read.

Supp Fig 18: Proteins are dynamic and domains can swing out of the way of each other. It is

possible the SRA domain could simply fold away from the DNA to allow the SBD and HNH to interact and cut – so a simple linear flat description does not tell the whole story.

Supp Fig 20: The band labeled as ~18 bp is running between the 20 and 25bp size standard, so is clearly NOT 18 bp (more like 22 bp). The ~50 bp fragment is clearly running faster than the 50bp size standard. Why report inaccurate lengths here?

-also note in the figure legend that vast majority of DNA molecules are completely uncut, so the chances of getting dual cut molecules is quite low, since the percentage of molecules cut even once is already quite small.

Point-by-point response to the reviewers

Reviewers' comments:

Reviewer #1 (Remarks to the Author):

Point 1. Liu et al. describe structural and biochemical characterization of a restriction enzyme ScoMcrA which is capable of specific cleavage of phosphorothioate DNA. The role and recognition of this DNA modification is a very interesting subject. Its studies can not only provide insights into the molecular recognition of nucleic acids but can also help build tools for nucleic acid processing/editing.

Response: We appreciate the reviewer's kind comments on our work!

Point 2. The results are presented clearly and the figures help understand the main points. This work is technically solid. The structures are refined to good parameters and the validation reports show satisfactory geometry of the models. The structure of SBD-DNA complex, which is the most important for the conclusions of the manuscript, is solved at relatively high resolution of 1.7 Å. The structural work is verified by mutagenesis studies. The main finding is that the sulphur atom of the phosphorothioate linkage is recognized by interactions with a hydrophobic pocket in the SBD domain. The generality of the proposed mechanism is verified for several SBD domains from various bacterial species. This is also demonstrated by a nice in vivo assay.

Response: We thank the reviewer very much for his/her appreciation of our work.

Point 3. One element which is missing in the manuscript is the structural analysis of the full length restrictase and in particular its potential interactions with the DNA substrate. A very general model of this interaction is shown in Suppl. Fig. 18., however a more detailed analysis needs to be done.

Response: According to the reviewer's nice suggestion, we have now added a discussion of the model of full-length ScoMcrA's potential interaction with PT-DNA as Supplementary Text. We have also added **Supplementary Fig. 23** and **Supplementary Fig. 24** to illustrate our point. This part of discussion is as follows:

“A model of full-length ScoMcrA dimer recognition and cleavage of PT-DNA

If we compare the structure of one protomer of full-length ScoMcrA and that of the SBD-SRA fragment in complex with PT-DNA, they can be superimposed onto each other very well, with a root-mean-square-deviation (RMSD) value of 0.453 Å for 263 atoms being compared (**Supplementary Fig. 23a**). However, if we compare the structures of dimeric full-length ScoMcrA (**Supplementary Fig. 23b**) and dimeric SBD-SRA fragment in complex with PT-DNA (**Supplementary Fig. 23c**), we find that their dimeric arrangements are different. A conformational change is needed for ScoMcrA to rearrange its dimeric assembly upon interaction with PT-DNA (**Supplementary Fig. 23d**).

Using the structure of full-length ScoMcrA and that of the SBD-SRA fragment in

complex with PT-DNA, we constructed a model of full-length ScoMcrA dimer in complex with PT-DNA (**Supplementary Fig. 24**). In this model, the residues of the SBD domain from one protomer responsible for phosphorothioate binding such as H116, R117, and P165 (magenta sticks in **Supplementary Fig. 24**), the active site residues H508, N522, and H531 of the HNH domain from the same ScoMcrA protomer (orange sticks in **Supplementary Fig. 24**), and the active site residues H508', N522', and H531' of the HNH' domain from the other ScoMcrA protomer (red sticks in **Supplementary Fig. 24**) are all accessible, and they can interact with PT-DNA at the same time in this dimeric configuration. The distance between the SBD-recognition site and the HNH'-contact site on PT-DNA is about 20 base pairs. Therefore, we propose that the full-length ScoMcrA dimer employs the SBD domain of one ScoMcrA protomer to recognize the sulfur atom as well as the G_{PT}GCC core motif, and uses the HNH and HNH' domains of both ScoMcrA protomers to cleave the two strands of PT-DNA ~20 base pairs away from the SBD-recognition site. This model is fully consistent with the cleavage occurred on the 5' side of PT link (**Supplementary Fig. 22**), and implies that the dimeric organization of ScoMcrA is essential for its biological activity.”

Point 4. The biological relevance of the dimeric structure of ScoMcrA alone shown in Fig 1c needs to be discussed.

Response: As suggested, we have now added a discussion of the model of full-length ScoMcrA's potential interaction with PT-DNA as the last paragraph of Supplementary Text. We have also added **Supplementary Fig. 24** to illustrate our point.

In this dimer, are the SBD residues responsible for phosphorothioate binding, the SRA residues potentially binding methylated DNA and the HNH residues forming the active site accessible? In other words, could they interact with DNA in this dimeric configuration?

Response: Yes, in the dimer, the SBD residues responsible for phosphorothioate binding and the active site residues of both HNH and HNH' domains are all accessible, and they could interact with DNA in this dimeric configuration (please see **Supplementary Fig. 24**). As predicted by the referee, one molecule of SRA domain in the model (showed in marine color in **Supplementary Fig. 24**) is accessible to be bound by the DNA. Potentially, the SRA domain recognizes methylated DNA and the HNH domains perform the cleavage on DNA.

What conformational changes (if any) would be required for the enzyme to bind and cleave the DNA?

Response: If we compare the structure of one Protomer of full-length ScoMcrA and that of the SBD-SRA fragment in complex with PT-DNA, they can be superimposed onto each other very well, with an RMSD value of 0.453 Å for 263 atoms being compared (**Supplementary Fig. 23a**). However, if we compare the structures of dimeric full-length ScoMcrA (**Supplementary Fig. 23b**) and dimeric SBD-SRA fragment in complex with PT-DNA (**Supplementary Fig. 23c**), we find that their

dimeric arrangements are different. A conformational change, such as a rotation of the SBD domain relative to the SRA domain, is needed for ScoMcrA to rearrange its dimeric assembly upon interaction with PT-DNA (black arrow in **Supplementary Fig. 23d**)

How does the arrangement of HNH domains in this dimer relate to the observed cleavage sites (Suppl Fig 20)?

Response: In our model, the residues of the SBD domain from one protomer responsible for phosphorothioate binding such as H116, R117, and P165, the key active site residues H508, N522, and H531 of the HNH domain from the same ScoMcrA protomer, as well as the active site residues H508', N522', and H531' of the HNH' domain from the other ScoMcrA protomer can interact with PT-DNA at the same time in this dimeric configuration (**Supplementary Fig. 24**). The distance between the SBD-recognition site and the HNH'-contact site on PT-DNA is about 20 base pairs, almost coincide with the distance from the PT link to either of the flanking cleavage sites in *in vitro* assay (**Supplementary Fig. 22**). Therefore, we propose that the full-length ScoMcrA dimer employs the SBD domain of one ScoMcrA protomer to recognize the sulfur atom in one DNA strand as well as the G_{PS}GCC core motif, and uses the HNH and HNH' domains of the two ScoMcrA protomers to cleave the two strands of the PT-DNA ~20 base pairs away from the SBD-recognition site. This model is fully consistent with our *in vitro* cleavage assay result, and implies that the dimeric organization of ScoMcrA is essential for its biological activity.

In the dimeric structure the two HNH are located close to each other. Does it imply that both will cleave on one side of the phosphorothioate and/or methyl modification?

Response: Yes, we agree with the reviewer that both HNH domains will cleave on one side of the phosphorothioate and/or methyl modification. This assumption is supported by DNA double strand cleavage on one side of the PT link when the hemi-phosphorothioated DNA was used (lane 3, 4 in Supplementary 22b).

Are there any other dimers that are generated by crystal contacts, the arrangement of which would fit better the cleavage data?

Response: All the molecules in the asymmetric unit of full-length ScoMcrA are shown in **Supplementary Fig. 2a**. These six molecules are assembled into three dimers: A-B pair, C-D pair, and E-F pair. There is no other rational dimeric arrangement which would fit the cleavage data.

How does the arrangement of SRA and SBD domains compare between the structure of full-length ScoMcrA and SBD-SRA fragment with DNA?

Response: The structure of a protomer of full-length ScoMcrA and that of a protomer of the SBD-SRA fragment in complex with PT-DNA can be superimposed onto each other very well, with an RMSD value of 0.453 Å for 263 atoms being compared (**Supplementary Fig. 23a**). However, if we compare the structures of dimeric full-length ScoMcrA (**Supplementary Fig. 23b**) and dimeric SBD-SRA fragment in

complex with PT-DNA (**Supplementary Fig. 23c**), we find that their dimeric arrangements are different. A rotation is needed for ScoMcrA to rearrange its dimeric assembly upon interaction with PT-DNA (**Supplementary Fig. 23d**).

Point 5. The crystallized DNA had two modifications but only one phosphorothioate interacts with the protein. However, E. coli and the M. morgani SBDs may form dimers on the DNA. Can two SBDs be modelled on one DNA without clashes? Additionally, in order to better understand the mechanism of SDB, the affinities (K_d) of SBD to doubly and singly PT-modified DNA need to be measured.

Response: As the reviewer suggested, we examined whether two ScoMcrA–SBD molecules can be modeled on one DNA, and we found that their SRGRR loops would clash with each other (**Supplementary Fig. 25a**). We have also measured the binding affinity (K_d) of ScoMcrA–SBD to doubly and singly PT-modified DNA, and the K_d values are very similar, with 1.469 μM for ScoMcrA–SBD binding to singly PT-modified DNA and 1.810 μM for ScoMcrA–SBD binding to doubly PT-modified DNA (**Supplementary Fig. 25b**).

However, we cannot exclude that SBD homologues from other bacteria (such as *E. coli* and *M. morgani*) can form dimers on the DNA as their loops corresponding to the ScoMcrA–SBD SRGRR loops are very much different. So it is possible that two sulfur atoms on two DNA strands are simultaneously associated by two SBD domains of the two homologues without steric clashes.

Point 6. In Suppl Figure 20a there are many more cuts indicated in panel (a) than bands observed in panel (b). How can this be explained? How was the DNA detected in this gel?

Response: The cuts presented in panel (a) are derived from sequencing of the cleavage products of a 118 bp 5'-radiolabeled PT-DNA cleaved by ScoMcrA, wherein multiple cleavages of variable efficiencies flanking PT-site DNA were detected (Figure 7, PLoS Genet. 2010 Dec 23;6(12):e1001253.). Each cleaved DNA fragment is denatured and resolved in the form of single strand in the sequencing gel. Those with free 5'-radiolabel were visualized by autoradiograph in a sensitive way. In this study, a 107 bp PT-DNA identical to the internal region of 118 bp duplex was cleaved by ScoMcrA (panel b), the cleaved products were visualized by ethidium bromide staining. Each of the bands in the gel contains multiple DNA fragments of approximately close size in this condition. The difference in the cuts between two assays is caused by utilization of two different techniques for DNA separation and visualization.

Point 7. Minor point: Labels in Suppl Fig 2 are swapped.

Response: Sorry for this carelessness. We have now swapped the figure legends for **Supplementary Fig. 2b** and **Supplementary Fig. 2c** in our revised manuscript.

Response to the Reviewer #2

Reviewer #2 (Remarks to the Author):

Point 1. The manuscript describes the crystal structure for a protein domain that binds phosphorothioate (PT) DNA. This domain is part of an enzyme, ScoMcrA, that recognizes the PT modification within the context of a conserved DNA sequence motif and cuts the modified DNA. A crystal structure at lower resolution is reported for the whole enzyme, while a high-resolution structure is reported for the domain responsible for binding the PT-modified DNA. This work contributes a structural understanding of how PT DNA can be recognized.

Response: We are grateful for the appreciation of the reviewer for our work!

Point 2. The writing needs a little polishing to improve the English and make it more readable.

There are a number of inconsistencies that require correction (for example, in the introduction the authors state there are ‘three consensus sequence patterns of PT modifications’, then proceed to list four such patterns in four different organisms).

Response: We are trying our best to polish our manuscript and improve the English. All changes of the texts are highlighted for your information. In the introduction of our revised manuscript, we have changed “three consensus sequence patterns of PT modifications” to “four consensus sequence patterns of PT modifications”.

Point 3. There are also major problems with the interpretation presented for the crystal data. For example, the authors state that a positive charge on an arginine residue (R190) forms hydrogen bonds to negatively charged O6 atoms of guanine (G3) and (G4), but also that this positive charge forms a hydrogen bond to the positively charged N4 atom of cytosine (C4). Two positive charges do not form hydrogen bonds to each other – this interpretation of the crystal data is clearly false, as it is simply not physically possible.

Response: Thanks for correcting our mistake! We have now deleted the statement of hydrogen bond between R190 and the N4 atom of C4' in the main article, and we have also removed the indication of hydrogen bond between R190 and the N4 atom of C4' in **Fig. 2d** and **Supplementary Fig. 11**.

The authors state that the ‘sulfur-recognizing residues’ they identify are highly conserved residues within PT-recognizing homologs they identify in sequence databases, yet one of these ‘highly conserved’ residues occurs only in their enzyme (R117), while the ‘gate’ residue Y164 is present in only 7 of the 23 homolog enzyme sequences presented in their alignment (Supp. Fig 15). These residues are clearly not highly conserved, as is claimed. It would seem important to at least comment on how the other residues at the Y164 position might or might not function in a similar manner to the proposed role of Y164.

Response: As advised, We have now made the following corrections on R117 and Y164 in the revised manuscript:

“Sequence alignment of these putative SBD domains shows that some of the

sulfur-recognizing residues of ScoMcrA–SBD such as P165 and H116 are the most highly conserved (**Fig. 5b**, **Supplementary Fig. 15a** and **Supplementary Table 3**). R117, the positively charged residue forming electrostatic interaction with the sulfur atom in PT-DNA, is replaced by a similarly positively charged lysine in most of the SBD homologues (**Supplementary Fig. 15a**). Presumably, these SBD homologues use lysine at this position to form electrostatic interaction with the PT-DNA sulfur.”

“Among the residues in the sulfur-binding pocket, Y164 is unique in that it plays two roles: employing its β -methylene group to form van der Waals interaction with the sulfur atom, and using its hydroxyl group to make hydrogen bonds with the N7 atom of G^{5'} and the N4 atom of C⁵. PT-DNA sequences in other bacteria possess consensus motifs other than G_{PS}GCC/G_{PS}GCC, therefore it is not a surprise for us to find that Y164 is only conserved in a subset of SBD homologues (7 out of the 23 SBD homologues in **Supplementary Fig. 15a**). Most likely, the SBD homologues in bacteria containing G_{PS}AAC/G_{PS}TTC, G_{PS}ATC/G_{PS}ATC, and C_{PS}CA core sequences employ residues different from tyrosine at this position to recognize these consensus PT-DNA sequences.”

In general, the interpretation of the crystal data would benefit from a more precise, accurate analysis, which I hope the authors will undertake and describe more accurately in a revised and improved manuscript.

Response: According to the review’s suggestion, we have now compared the dimeric arrangement of full length ScoMcrA structure with that of SBD-SRA in complex with PT-DNA (**Supplementary Fig. 23**). In addition, we have made a model of full length ScoMcrA dimer recognizing PT-DNA for cleavage (**Supplementary Fig. 24**), and compared it with our experimental *in vitro* PT-DNA cleavage results. Furthermore, we analyzed the possibility of two SBD binding on the same PT-DNA by both structural analysis and *in vitro* binding assay using purified proteins (**Supplementary Fig. 25**). We have also made point mutations on ScoMcrA–SBD and tested the effects of these mutations on the association between ScoMcrA–SBD and PT-DNA (**Supplementary Fig. 15b**), in order to corroborate our structural observations. We have added according explanations at relevant places in the main text, figure legends, and supplementary text. All the changes have been highlighted in the revised manuscript. We hope that we have made the interpretation of the crystal data precise and accurate by careful editing of the manuscript.

Point 4. The topic of how PT modification can be specifically bound is important and interesting, and is well worthy of publication. The presentation and analysis just needs to be tightened up to generate an excellent paper.

Response: We are very much encouraged by the reviewer’s nice comment. We have made thorough editing of the manuscript by the team work, and all revisions are highlighted in the texts. We hope these changes make the paper presented in a concise and accurate way.

Point 5. Specific comments:

Introduction:

Page 3: State ‘three consensus patterns’, then list 4 patterns – need to be consistent.

Response: Sorry for this careless mistake. We have corrected “three consensus patterns” to “four consensus patterns” in our revised manuscript.

Point 6. Page 5: states that the phosphorothioate sulfur is negatively charged and forms ‘salt bridges’ with R117, but then the paper argues this sulfur is bound in a hydrophobic uncharged pocket. How is this consistent (negatively charged and hydrophobic)? This needs an explanation of why a sulfur atom behaves differently than an oxygen atom and how any differences can lead to specific recognition of the sulfur in PT DNA.

Response: This is a key question for this manuscript. To make this point clearer. We have added the following explanation as supplementary text in our revised manuscript.

“The electronic structure of phosphorothioate has been well characterized in the previous literatures (Scientific Reports, 2017, 7:42823; J. Phys. Chem. B 2012, 116:10639-10648). In unbound phosphorothioated DNA, the negative charge favors the sulfur atom in an anionic form [$\text{P}(=\text{O})\text{S}^-$], similar to the phosphate group in normal DNA (see Figure 6 in Scientific Reports, 2017). On the other hand, the amount of electron density of the sulfur atom is distributed in a much larger region in phosphorothioate group than that of the oxygen atom in phosphate group. Owing to a larger radius and higher polarizability, the sulfur atom favors a larger London dispersion force and less electrostatic interaction. SBD protein provides two interacting sites, one from R117 (positively charged, salt-bridge hard partner) and another from hydrophobic cavity (uncharged, van der Waals interaction dominantly, soft partner). Phosphorothioate binds R117 with the hard P-O side and hydrophobic cavity with the soft P-S side. Therefore, we argue that the sulfur atom of PT-DNA is bound in a hydrophobic uncharged cavity by dispersion force, in a comparative viewpoint.

Sulfur and oxygen belong to the same group (group VIA) in the periodic table. The outmost shell electron configuration of oxygen is $2s^22p^4$, and the outmost shell electron configuration of sulfur is $3s^23p^4$. They both tend to attract two electrons to make their outmost shell electron configuration become ns^2np^6 ($2s^22p^6$ for oxygen and $3s^23p^6$ for sulfur, respectively). When oxygen forms a covalent bond with phosphorus in normal DNA, it would attract one electron from the phosphorus atom and one electron from the counter-cation (usually Na^+ in physiological condition) to become negatively charged. Sulfur does the same thing. When sulfur forms a covalent bond with phosphorus in PT-DNA, it would attract one electron from the phosphorus atom and one electron from the counter-cation. However, sulfur is much bigger than oxygen. The atomic radius of oxygen is estimated to be 48 picometer while the atomic radius of sulfur is estimated to be 88 picometer. Hence the electronegativity of sulfur is much less than that of oxygen, being 2.58 compared to 3.44 for that of oxygen by the Pauling scale. Therefore, sulfur in PT-DNA carries much less negative charge than oxygen in normal DNA, although it still does carry negative charge and thus can form

a salt bridge with the positively charged R117 of ScoMcrA.

Hydrophobicity/hydrophilicity (hydro- means water) is determined by whether an atom (or a molecule) can form hydrogen bond with water or not. Oxygen is highly electronegative, and thus it carries enough negative charge to form hydrogen bonds with water, and behaves as hydrophilic. In contrast, sulfur is not that electronegative, and it does not possess enough negative charge to form hydrogen bonds with water. Therefore, it behaves as hydrophobic. For example, serine is hydrophilic whereas cysteine (with one oxygen-to-sulfur replacement compared with serine) is much more hydrophobic than serine, and methionine (which contains a sulfur atom) is certainly hydrophobic.

In summary, compared with oxygen in normal DNA, the sulfur atom in PT-DNA is much more hydrophobic. At the same time, it is negatively charged, although the negative charge it carries is much less than that of oxygen. Therefore, it is no surprise that ScoMcrA-SBD employs a hydrophobic pocket to selectively recognize the sulfur atom in PT-DNA, but not the oxygen atom in normal DNA. At the same time, the sulfur atom in PT-DNA is still negatively charged and can form salt bridge with strongly positively charged residues such as R117.”

Results:

Point 7. Page 9: would R117 form a similar salt bridge with oxygen rather than sulfur? Is there anything here that would be specific for sulfur over the normal oxygen?

Response: Yes, in the case of normal DNA, R117 would also form a salt bridge with oxygen. Actually, the electro-negativity of oxygen is higher than sulfur, therefore oxygen carries more negative charge than sulfur. Hence, with respect to salt bridge with R117, there is nothing specific for sulfur over the normal oxygen.

Point 8. Page 10: Mutation to Proline165: mutating a proline to any other amino acid will change the backbone of the protein, as no other amino acid has the same backbone connectivity as Proline, so best to be careful in interpreting the resulting loss of activity (could be due to improper folding/shape rather than simply charge/no charge).

Response: Thanks for the nice suggestion, and we have now added this caution in our revised manuscript (see underlined words):

“It was confirmed that mutation of P165 or R117 substantially disrupted the ability of ScoMcrA-SBD to bind PT-DNA, while mutation of other residues such as H116 also diminished the association to varying degrees (**Supplementary Fig. 10; no other amino acid has the same backbone connectivity as proline, so mutating a proline to any other amino acid will change the backbone of the SBD protein and might lead to loss of its activity because of improper folding instead of changing of hydrophobicity**). These EMSA and fluorescence polarization assay results indicate that both hydrophobic and electrostatic interactions, as well as correct folding of the SBD domain, play crucial roles for the sulfur-binding cavity of ScoMcrA-SBD to recognize the sulfur atom on PT-DNA.”

Point 9. Page 10: Why not try the R117K mutation that is common in the homologs?

Response: As suggested, we made R117K mutation of ScoMcrA–SBD, and measured its binding affinity with PT-DNA using the fluorescence polarization assay. Its dissociation constant (K_d) was measured to be 1550 nM, not much different from the K_d value of wild-type ScoMcrA–SBD for PT-DNA, which was 1317 nM (**Supplementary Fig. 15b**). The R117K mutant may employ the positively charged side-chain of lysine to form electrostatic interaction with the negatively charged phosphorothioate of PT-DNA.

Point 10. Page 10: Note that A168 is not conserved in other homologs (most have a large, charged amino acid (His or Arg) at this position. The effect of removing the methyl group (A168G) is small. Either homologs have quite different binding pockets, or the role of the residue at this position is not simply a van der Waals interaction, or perhaps is not so important to PT binding as is argued here.

Response: We agree with the comments on the importance of A168. We have now removed the sentence describing A168G mutation in the text, and added another sentence in our revised manuscript as advised:

“A168 is not as conserved as some other residues such as P165 in ScoMcrA–SBD homologues (see **Fig. 5b** below), therefore its importance for PT-DNA binding might simply provide a van der Waals interaction.”

Point 11. Page 11: Does the reported hydrogen bond of Y164 to the N4 of C5 contribute to opening the sulfur binding pocket and keeping the pocket open? Seems this may be important for PT recognition, and might explain why Y164 is not well conserved in homologs, as these might recognize PT DNA in a different sequence motif context.

Response: Very nice comment with respect to the role of Y164! We have incorporated the reviewer’s suggestion into our revised manuscript as follows:

“The hydrogen bonds of Y164 to the N4 of C⁵ and the N4 of C⁶ atoms might contribute to flipping the hydroxyphenyl group of Y164 and opening the sulfur-binding pocket of ScoMcrA–SBD. The importance of Y164 for the recognition of the G_{PS}GCC core sequence is consistent with the fact that it is not well conserved in ScoMcrA–SBD homologues as different PT-DNA core sequence motifs would require different corresponding residues for binding at this position.”

Point 12. Page 12: The lack of electron density for Y164 when not in complex with PT DNA indicates this residue is flexible and is sampling multiple configurations, not that it is in a closed state – only 2 of 6 molecules have such a closed state.

Response: To pinpoint the configurations of Y164 in six ScoMcrA molecules, we have changed this part to “When not in complex with PT-DNA, Y164 is flexible and samples a variety of conformations. Of the six ScoMcrA molecules in the asymmetric unit, only two have clearly observable electron densities for the side-chain of Y164 and both of which are in the “closed” state (**Supplementary Fig. 13**). In these two molecules, the hydrophobic phenyl ring of the side-chain of Y164 covers the opening

of the non-polar sulfur-binding cavity (**Fig. 3a** and **Supplementary Fig. 14a**). The other four ScoMcrA molecules lack clearly observable electron densities for the side-chain of Y164 (**Supplementary Fig. 13**.)” in our revised manuscript.

Point 13. Page 13: It may be that binding to the GGCC motif causes Y164 to rotate into the ‘open’ position, not necessarily due to binding to the sulfur of PT DNA as is implied.

Response: As suggested, we have made the correction in our manuscript: “This conformational change of ScoMcrA–Y164 could be caused by the binding of ScoMcrA–SBD either to the sulfur atom of PT-DNA or the binding to the GGCC core motif, or both interactions may make contributions.”

Point 14. Page 13: what axis does the sulfur rotate about?

Response: The sulfur atom rotates about the phosphodiester backbone of PT-DNA. We have made the corresponding correction in our manuscript:

“When compared with the DNA strand that is not in contact with ScoMcrA–SBD, the sulfur atom on the SBD-bound strand of PT-DNA is rotated about the phosphodiester backbone of PT-DNA by 80° outward so that it can be more comfortably fit into the sulfur-binding cavity on ScoMcrA–SBD (**Fig. 4a**).”

Point 15. Page 14: R117 and Y164 are not highly conserved in the alignment shown in Sup Fig 15! It appears H116, P118, P165, F166, W167, L169, W175 are highly conserved: seems to be too much focus on R117 and Y164, and not enough attention to the other structural elements (P118, F166, W167, L169, W175).

Response: Actually we had a discussion on the roles of P118, F166, W167, L169, W175 included as supplementary text in our previous manuscript. We have now moved this paragraph to the main article upon the reviewer’s suggestion, which is as follows:

“Besides these residues, there are other highly conserved residues of ScoMcrA–SBD, including P118, V119, L120, L121, F166, W167, L169, and W175 (**Fig. 5b** and **Supplementary Fig. 15a**). These residues do not contribute to the interaction between ScoMcrA–SBD and PT-DNA, but participate in forming the hydrophobic core of the ScoMcrA–SBD domain (**Supplementary Fig. 16**). Presumably, mutation of these residues would compromise the folding of the ScoMcrA–SBD domain, which accounts for their high conservation among homologues.”

Point 16. Page 15: How were the three expressed homologs “distant” in relationship to ScoMcrA?

Response: ScoMcrA is originated from *Streptomyces coelicolor*, which belongs to the phylum of actinobacteria. The three ScoMcrA homologues we tested in our manuscript are from *Escherichia coli*, *Morganella morganii*, and *Streptomyces gancidicus*. Both *Escherichia coli* and *Morganella morganii* belong to the phylum of proteobacteria, which is a separate phylum different from the actinobacteria phylum.

Therefore, they are considered as “distant” homologues of ScoMcrA. *Streptomyces gancidicus* is much closer to *Streptomyces coelicolor*, and we have now made corresponding correction in our manuscript, as follows:

“To investigate whether these ScoMcrA–SBD homologues could indeed associate with PT-DNA, we selected three SBD homologues from *Streptomyces gancidicus*, *Escherichia coli*, and *Morganella morganii*. ScoMcrA originates from *Streptomyces coelicolor*, which belongs to the phylum of actinobacteria. On the other hand, both *Escherichia coli* and *Morganella morganii* belong to the phylum of proteobacteria, which is a separate phylum different from the actinobacteria phylum. Therefore, the *Escherichia coli* and *Morganella morganii* homologues are distantly related to ScoMcrA. These three ScoMcrA–SBD homologues were heterologously expressed in the *E. coli* strain BL21(DE3), purified, and were examined for their abilities to interact with PT-DNA by the EMSA and fluorescence polarization assays *in vitro*.”.

Point 17. Page 15: If the expressed homologs were distant, why did they recognize all three sequence motif contexts (GGCC, GATC, GAAC)? Do these have an SRGRR loop or equivalent sequence recognition element?

Response: Our opinion is that the *Streptomyces gancidicus*, *Escherichia coli*, and *Morganella morganii* ScoMcrA–SBD homologues might not have stringent requirement for particular PT-DNA sequences for binding. ScoMcrA–SBD possesses an SRGRR loop, which specifically recognizes the G_{PS}GCC sequence. On the other hand, the *Streptomyces gancidicus*, *Escherichia coli*, and *Morganella morganii* ScoMcrA–SBD homologues do not have an SRGRR loop although each of them has a loop at the equivalent position. We have determined the crystal structure of *Streptomyces pristinaespiralis* SBD homologue in complex with G_{PS}GCC and with G_{PS}AAC PT-DNA sequences. Our structures show that indeed *Streptomyces pristinaespiralis* SBD’s loop corresponding to the SRGRR loop does not play important roles in binding to PT-DNA core sequence motifs, therefore it has a loose requirement for PT-DNA binding partners and can bind to a variety of sequences (manuscript in preparation). It might be the same scenario for *Streptomyces gancidicus*, *Escherichia coli*, and *Morganella morganii* ScoMcrA–SBD homologues. From this point, the SRGRR loop in ScoMcrA is somehow unique as it interacts with the moieties of DNA bases.

Point 18. Page 20: HNH endonucleases are efficient – they are not inherently poor at DNA cutting, and PD-ExK endonucleases are not inherently better. This suggestion is unlikely to succeed.

Response: Thanks for pointing out this mistake. we have deleted this sentence in our revised manuscript.

Point 19. Page 20: There has not been a demonstration of sequence-context independent recognition of PT DNA made, so the suggestion to broadly characterize PT motif using S. gancidicus enzyme is premature. If S. gancidicus is found to be highly flexible in recognition, what is the basis of that flexibility (compared to

ScoMcrA)? Does it lack a DNA recognition loop analogous to SRGRR of *ScoMcrA*?

Response: We have performed the fluorescence polarization assay to measure the binding affinities of the *S. gancidicus* SBD homologue and *ScoMcrA*–SBD for PT-DNA with core sequences of G_{PS}GCC, G_{PS}AAC, or G_{PS}ATC (**Supplementary Fig. 21a, b, c**). *S. gancidicus* SBD homologue was found to associate with all three kinds of PT-DNA, whereas *ScoMcrA*–SBD only interacted with G_{PS}GCC and did not bind G_{PS}AAC or G_{PS}ATC. Therefore, *S. gancidicus* SBD homologue is indeed highly flexible in recognition of PT-DNA with different core sequences.

As for the basis of the flexibility of the *S. gancidicus* SBD homologue, we have aligned and compared the protein sequences of *ScoMcrA*–SBD and the *S. gancidicus* SBD homologue (**Supplementary Fig. 21d**). We find that *ScoMcrA*–SBD possesses several negatively charged residues, E156, D157, and D160, which are absent in the *S. gancidicus* SBD homologue. On the other hand, the *S. gancidicus* SBD homologue possesses a positively charged residue, K191, which is absent in *ScoMcrA*–SBD. We hypothesize that the negatively charged residues, E156, D157, and D160 in *ScoMcrA*–SBD resulted in repulsion with the negatively charged PT-DNA, so that *ScoMcrA*–SBD has decreased binding affinities with PT-DNA harboring the G_{PS}GCC core sequence and has lost interaction with PT-DNA containing the G_{PS}AAC or G_{PS}ATC core sequences. In contrast, the positively charged residue K191 present in the *S. gancidicus* SBD homologue resulted in increased binding affinities with PT-DNA harboring all three kinds of core sequences.

To verify our hypothesis, we have made mutations in *ScoMcrA*–SBD replacing negatively charged E156, D157, and D160 with positively charged residues and it indeed enhanced the association between *ScoMcrA*–SBD and PT-DNA. We are now in the process of organizing this part of data and preparing another manuscript, together with the data of crystal structures of *S. pristinaespiralis* SBD homologue in complexes with G_{PS}GCC and with G_{PS}AAC core PT-DNA sequences (see the response to *Point 17*).

Point 20. Page 20: Agree that PT DNA is widespread and PT-readers are important in processing the modification information.

Response: We appreciate the reviewer’s understanding of our results and conclusion that PT-DNA is widespread and PT-DNA readers are important in processing the modification information.

Point 21. Figure 2: suggest showing the guanine O6 and N7 and the cytosine N4 positions in the figure (as bumps on the bases) and the contacts to these by the various amino acid residues.

Response: We have revised **Fig. 2** as suggested by the reviewer.

Point 22. Why are there no contacts to the G6:C6 base pair, which is part of the GGCC specific motif? Seems there should be something specifying recognition for this base pair.

Response: In page 12 of our revised manuscript, we mentioned that “the N4 atom of

C⁶ forms a weak hydrogen bond with the hydroxyl group of Y164 and an electrostatic interaction with the carboxyl group of D160.” We have now made correction on **Fig. 2c** to add interactions with the N4 atom of C⁶.

Point 23. Figure 5: Suggest using residues observed in SBD homolog enzymes to guide mutations made for testing ScoMcrA activity, in addition to mutations expected to K/O the enzyme function (such as R117 to Q or K; Y164 to M; A168 to H or R).

Response: As suggested, we have made mutations of the R117K, R117Q, Y164M, A168H, and A168R to ScoMcrA–SBD, and measured their binding affinities for PT-DNA using the fluorescence polarization assay (**Supplementary Fig. 15b**). Point mutation of R117K did not appreciably affect the binding affinity between ScoMcrA–SBD and PT-DNA, while the R117Q mutation decreased the association between ScoMcrA–SBD and PT-DNA presumably because glutamine is not as positively charged as arginine and lysine. Interestingly, the Y164M mutation strengthened the binding between ScoMcrA–SBD and PT-DNA, presumably because the hydrophobicity of methionine is not less than that of tyrosine, and methionine also possesses substantial conformational malleability. A168 contributes to the association with PT-DNA by using its β -methyl group to make hydrophobic interaction with the sulfur atom of PT-DNA. Other amino acids can replace alanine at this position since they possess β -methylene groups which can function in the same way. Therefore, it is not surprised to find that mutation of A168 to other residues such as A168H or A168R in ScoMcrA–SBD did not substantially decrease its binding affinity for PT-DNA (**Supplementary Fig. 15b**).

We summarized our results in **Supplementary Fig. 15**, and described and explained the mutation effects on the binding affinity to PT DNA in line 318-343, Pages 15 and 16.

Point 24. Supp Fig 11 and 12: The DNA does not seem to be base paired in these figures: is this the case? If so, comment on why it is not base paired; otherwise correct the image.

Response: According to the comment, we have now re-made the image, and viewed the base pairing of DNA as well as the interaction between ScoMcrA and PT-DNA from another angle. The DNA is base paired in this figure. To make it clearer, we indicated the hydrogen bonding in DNA base pairs by the orange dashed lines, G4 is paired with C4', and C5 is paired with G5'. C3', the base which is paired with G3, is not shown for clarity.

Point 25. Note: there cannot be a H-bond from positive Arg to a positive N4-cytosine group. The contact pattern here needs to be re-analyzed and corrected.

Response: Thanks for pointing out this error. We have now deleted the description of hydrogen bond between R190 and the N4 atom of C4' in the main article, and we have also removed the indication of hydrogen bond between R190 and the N4 atom of C4' in **Fig. 2d** and **Supplementary Fig. 11**.

Point 26. Supp Fig 13 and 14: Perhaps comment on the observation that Y164 is dynamic, not fixed into a single position (4 of 6 chains in the crystal have flexible Y164, ie no density).

Response: We have revised the figure legends to **Supplementary Fig. 13** and **Supplementary Fig. 14** according to the reviewer's very nice suggestion, see the following underlined words:

“Supplementary Figure 13 Comparison of the conformations of Y164 in the six different molecules of ScoMcrA in the asymmetric unit of the crystal structure of full-length ScoMcrA by itself. Only in chain C and chain F was the electron density of Y164 clear enough to unambiguously observe the hydroxyphenyl groups of Y164, which are both in the “closed” conformation. The hydrophobic phenyl ring of the side-chain of Y164 covers the opening of the non-polar sulfur-binding cavity. In chains A, B, D, and E, the electron densities of the side-chain of Y164 were not clear, and Y164 was modeled as alanines in these chains. This observation indicates that Y164 of ScoMcrA is flexible and samples a variety of conformations when ScoMcrA is not in complex with PT-DNA.

Supplementary Figure 14 Front views for the “closed” state of Y164 in the PT-DNA–unbound ScoMcrA-SBD and for the “open” state of Y164 in the PT-DNA–bound ScoMcrA-SBD. This suggests that when not in complex with PT-DNA, Y164 of ScoMcrA-SBD is flexible and samples a variety of conformations, including the “closed” state which the hydroxyphenyl ring of Y164 covers the opening of the sulfur-binding pocket. On the other hand, binding to PT-DNA (both to the sulfur atom and to the G_{PT}GCC core sequence) induces the flipping of hydroxyphenyl ring of Y164 and stabilizes it in the “open” state. (a) In the PT-DNA–unbound ScoMcrA-SBD, Y164 exists in the “closed” state, and its phenyl ring covers the sulfur-binding cavity of ScoMcrA-SBD. The electrostatic surface of ScoMcrA-SBD is shown. Y164 is displayed both in stick and space-filling representations. (b) In the PT-DNA–bound ScoMcrA-SBD, Y164 exists in the “open” state. The hydroxyphenyl group of its side-chain rotates away to allow the sulfur atom of PT-DNA to access the sulfur-binding cavity of ScoMcrA-SBD. The sulfur atom of PT-DNA is shown as a green sphere. The rest of PT-DNA is omitted for clarity.”

Point 27. Supp Fig 17: Please make text under the highlights BOLD to make it easier to read.

Response: We have now made correction on the **Supplementary Fig. 17**, and made the text under the highlights BOLD according to the reviewer's suggestion.

Point 28. Supp Fig 18: Proteins are dynamic and domains can swing out of the way of each other. It is possible the SRA domain could simply fold away from the DNA to allow the SBD and HNH to interact and cut – so a simple linear flat description does not tell the whole story.

Response: We agree with the reviewer that proteins are dynamic and flexible. However, in our case of the ScoMcrA protein, the spatial positions of the four domains, head, SBD, and SRA, and HNH arrange in a linear manner. The loops

connecting adjacent domains are relatively short. The loop connecting SBD and SRA domains is only five residues long, and the loop connecting SRA and HNH domains is only 18 residues long. There is only limited flexibility between adjacent domains (see **Supplementary Fig. 4**). We would think that it is highly unlikely that a dramatic conformational change that allows domains to completely swing out (for example, the SRA folds away to allow SBD and HNH to interact with each other) would happen.

Point 29. Supp Fig 20: The band labeled as ~18 bp is running between the 20 and 25bp size standard, so is clearly NOT 18 bp (more like 22 bp). The ~50 bp fragment is clearly running faster than the 50bp size standard. Why report inaccurate lengths here?

-also note in the figure legend that vast majority of DNA molecules are completely uncut, so the chances of getting dual cut molecules is quite low, since the percentage of molecules cut even once is already quite small.

Response: Yes, you are right that the length for the two fragments are inaccurate. Sorry for this carelessness. We have changed the length to 21 bp and 47 bp, respectively, based on previous report and the result shown in revised **Supplementary Fig. 22b**. The cuts presented in panel a are derived from sequencing of the cleavage products of a 118 bp 5'-radiolabeled PT-DNA by ScoMcrA, wherein multiple cleavages of variable efficiencies flanking PT-site DNA were detected (Figure 7, PLoS Genet. 2010 Dec 23;6(12):e1001253.). Therefore, each band in the gel (panel b) contains several DNA fragments of close length. This is one reason we report wrong length for the bands. Given the cleavage flexibility of ScoMcrA, we use the cleavage site with the highest cut frequency (panel a, rectangle in dashed line) to depict the location of cleavage as well as the length of the cleaved fragments with respect to the PT link or to the end. Thus the band of 21 bp (you suggested 22 bp) corresponds to 23 nt/19 nt (panel a), the band of 47 bp is 46 nt/49 nt (panel a). Related revisions are made in this regard accordingly.

Your explanation for dual cuts is very useful to help readers to understand the result, and we have also included them in the revised **Supplementary Text** with the subtitle of “**Cleavage by the ScoMcrA HNH domain occurs at a distance of ~23 bp away from the phosphorothioate linkage at the 5' side of G_{PS}GCC**”.

REVIEWERS' COMMENTS:

Reviewer #1 (Remarks to the Author):

I read the authors' rebuttal carefully and I conclude that they addressed the comments of the referees appropriately.

This manuscript would benefit from English language copyediting/proofreading.

Reviewer #2 (Remarks to the Author):

The authors have largely addressed the points raised in the initial review.

One minor point: It still seems incorrect to term a protein a "distant homolog" (pg 19, line 376) based on the organism in which it is found, rather than on the protein sequence of the enzyme. Bacteria can experience lateral gene transfer even between quite different phyla, so that distantly related bacteria may have nearly identical individual proteins. Restriction-Modification enzymes are particularly mobile in lateral gene transfer, so it is not assured that ScoMcrA homologs from 'distant' bacterial phyla are actually 'distantly' related. Better to go by protein sequence similarity to determine how closely or distantly related these homologs might be.

The PDB coordinates for the structure would be helpful to include in the abstract (or at least at the very beginning of the results).

Reviewer #1 (Remarks to the Author):

I read the authors' rebuttal carefully and I conclude that they addressed the comments of the referees appropriately.

This manuscript would benefit from English language copyediting/proofreading.

The main text was considerably edited by the editor of Nature Communications, Dr. Karin Kuehnel. I am deeply grateful to her help. Also this manuscript has been carefully edited by Dr. Neil Price in the initial submission to Nature Communications. After receiving the decision letter, he revised the abstract again based on the version by Dr. Karin Kuehnel. Dr. Price is an expert in microbiology and biochemistry at National Center for Agricultural Utilization Research, USDA, Peoria, IL. You may contact him by E-mail: Neil.Price@ARS.USDA.GOV.

Reviewer #2 (Remarks to the Author):

The authors have largely addressed the points raised in the initial review.

One minor point: It still seems incorrect to term a protein a "distant homolog" (pg 19, line 376) based on the organism in which it is found, rather than on the protein sequence of

the enzyme. Bacteria can experience lateral gene transfer even between quite different phyla, so that distantly related bacteria may have nearly identical individual proteins. Restriction-Modification enzymes are particularly mobile in lateral gene transfer, so it is not assured that ScoMcrA homologs from 'distant' bacterial phyla are actually 'distantly' related. Better to go by protein sequence similarity to determine how closely or distantly related these homologs might be.

I agree with the reviewer that the genes could be acquired through horizontal transfer from other bacterium of distantly related phyla, particularly for those encoding restriction endonuclease and DNA methylase. Therefore, the phylogenetic relationship among bacteria cannot be applied to that of proteins from these compared hosts. For this concern, we constructed another tree based on the homology of amino acid sequences of 2761 SBD-containing homologues, the tree was summarized as the Supplementary Figure 17, from which we can see that SBD homologs from *Escherichia coli* and *Morganella morganii* are indeed distant relatives to ScoMcrA.

The order for the supplementary figures after 17 is adjusted accordingly in the main text and supplementary information.

The PDB coordinates for the structure would be helpful to include in the abstract (or at least at the very beginning of the results).

The PDB codes for three protein structures are released from PDB. The PDB code was given when each appeared for the first time, and as required, a URL link was added for each PDB code throughout the manuscript.